# Evaluating LLM-Contaminated Crowdsourcing Data Without Ground Truth

**Yichi Zhang**[*]
DIMACS, Rutgers University
yz1636@dimacs.rutgers.edu

**Jinlong Pang**[*]
University of California, Santa Cruz
jpang14@ucsc.edu

**Zhaowei Zhu**
University of California, Santa Cruz
zwzhu@ucsc.edu

**Yang Liu**
University of California, Santa Cruz
yangliu@ucsc.edu

## Abstract

The recent success of generative AI highlights the crucial role of high-quality human feedback in building trustworthy AI systems. However, the increasing use of large language models (LLMs) by crowdsourcing workers poses a significant challenge: datasets intended to reflect human input may be compromised by LLM-generated responses. Existing LLM detection approaches often rely on high-dimensional training data such as text, making them unsuitable for structured annotation tasks like multiple-choice labeling. In this work, we investigate the potential of peer prediction—a mechanism that evaluates the information within workers' responses—to mitigate LLM-assisted cheating in crowdsourcing with a focus on annotation tasks. Our approach quantifies the correlations between worker answers while conditioning on (a subset of) LLM-generated labels available to the requester. Building on prior research, we propose a training-free scoring mechanism with theoretical guarantees under a novel model that accounts for LLM collusion. We establish conditions under which our method is effective and empirically demonstrate its robustness in detecting low-effort cheating on real-world crowdsourcing datasets.

## 1 Introduction

High-quality human feedback plays a crucial role in shaping modern AI systems, such as ChatGPT, to be trustworthy, safe, and aligned with human preference Ouyang et al. [2022]. This feedback is usually collected through crowdsourcing, where a requester posts tasks and recruits trained experts or general participants to complete them. For example, platforms like Amazon Mechanical Turk and Prolific are widely used by researchers for tasks such as data labeling and survey administration.

However, the integrity of this data collection process is increasingly threatened by the very AI models it seeks to improve. Evidence suggests that many agents rely on large language models (LLMs) to complete tasks such as abstract summarization [Veselovsky et al., 2023] and peer review [Liang et al., 2024], leading to datasets that no longer reflect genuine human input, a phenomenon known as *LLM contamination*. While LLM-assisted annotations or responses may sometimes be more accurate or coherent than human's [Gilardi et al., 2023, Törnberg, 2023, Kaikaus et al., 2023], their widespread use undermines the fundamental goal of gathering diverse and authentic human opinions. This issue is particularly concerning in applications that depend on human subjectivity or expertise, such as preference labeling and toxicity detection, where human judgment remains the gold standard.

---

[*]Equal contribution.

*How to detect potential LLM-misuse and evaluate the quality of a human-contributed dataset?*
Existing black-box methods for detecting LLM-generated content primarily focus on distinguishing human-written text from AI-generated text by analyzing statistical differences in writing style [Gehrmann et al., 2019, Mao et al., 2024, Koike et al., 2024, Hu et al., 2023]. While effective in open-ended text generation tasks, these training-based approaches are not well-suited for structured annotation tasks like multiple-choice labeling, where the discrete responses lack the rich textual features needed for classification.

In this paper, we develop a **theoretically robust scoring metric** to evaluate the quality of human contributions in addition to existing LLM answers, effectively distinguishing workers with less informative responses. Our method builds on *peer prediction*, a scoring mechanism designed to incentivize truthful reporting in the absence of ground truth Miller et al. [2005], Shnayder et al. [2016]. Traditional peer prediction methods evaluate a worker based on the correlation between her responses and those of her peers. In the standard model, every worker independently observes only high-effort signals, denoted as $X_i$ for worker $i$, after working on a set of i.i.d. questions. Peer prediction mechanisms are shown to be *information monotone*, meaning that workers maximize their expected score by truthfully reporting their high-effort signals.

However, these methods are ineffective when workers can coordinate on cheap signals, denoted as $Z_i$ for worker $i$, that are highly correlated and require minimal effort to produce. In our settings, LLM responses serve as such cheap signals. To address this limitation, we extend peer prediction by measuring correlations between workers' responses while conditioning on a signal $Z$ that is observable to the requester. For example, $Z_i$ and $Z$ could represent responses to the questions independently generated by the same LLM or different LLMs. Intuitively, by conditioning on $Z$, our method scores the information within workers' responses beyond what can be obtained from LLMs alone, ensuring that human effort is properly rewarded while AI-reliant is penalized.

Theoretically, we identify the conditions under which our mechanism maintains information monotonicity under two regimes: (1) for any form of manipulation strategy and (2) for a subset of practical cheating strategies, which we refer to as *lazy-reporting* strategies. A lazy-reporting strategy, for instance, includes blindly relying on LLM on a fraction of the tasks while answering the rest truthfully. To achieve information monotonicity in the general case, for an arbitrary pair of workers, $i$'s cheap signal $Z_i$ has to be approximately uncorrelated with $j$'s high-effort signal $X_j$ and their cheap signal $Z_j$, conditioned on $Z$ (Assumptions 4.1 and 4.2). For lazy-reporting strategies, a weaker condition suffices: conditioning on $Z$, the correlation between high-effort signals $(X_i, X_j)$ is stronger than the correlation between both $(Z_i, X_j)$ and $(Z_i, Z_j)$ (Assumption 4.5).

Empirically, we test our assumptions using two subjective labeling datasets and five types of commonly used LLMs. Our findings show that human responses consistently exhibit stronger correlations than LLM-generated responses when samples of $Z$ and $Z_i$ originate from the same model, supporting the validity of Assumption 4.5. However, the correlations between independent samples of LLMs responses are usually non-zero, suggesting that Assumptions 4.1 and 4.2 are too strong to hold in practice. Finally, under scenarios where Assumption 4.5 holds, we evaluate the effectiveness of our method in detecting low-effort agents. Echoing our theoretical insights, we show that our approach has the most robust performance on mixed crowd compared with all the baselines.

## 2   Related Work

Our study relates to the studies on LLMs usage in crowdsourcing. Cegin et al. [2023] find that ChatGPT can partially replace human workers in paraphrase generation, and Kaikaus et al. [2023] report that GPT labels are more consistent than humans in sentiment annotations. Ashktorab et al. [2021] demonstrate that "batch labeling", a framework of using AI to assist human labeling by allowing a single labeling action to apply to multiple records, can increase the overall labeling accuracy and increase the speed. However, several studies also highlight potential downsides: del Rio-Chanona et al. [2023] report a 16% decline in Stack Overflow activity following the rise of LLMs, and Ashwin et al. [2023] find that LLMs exhibit greater bias than human annotators in qualitative analysis. In this context, our paper introduces a scoring mechanism that quantitatively measures the informativeness of human reports relative to the information already provided by AI.

Our work builds on advancements in information elicitation, particularly peer prediction. As a generalization of the correlated agreement (CA) mechanism [Shnayder et al., 2016], our method

can score the quality of human responses conditioned on an external information source. We chose the CA mechanism for its simplicity and robust empirical performance [Burrell and Schoenebeck, 2021, Zhang and Schoenebeck, 2023a]. Our approach is also inspired by Kong and Schoenebeck [2018], who proposed using conditioned mutual information to elicit hierarchical information. Their framework assumes an information hierarchy where more effort yields strictly more informative signals, so low-effort signals can be elicited first, which are used to further elicit high-effort signals. In contrast, our method assumes that the principal has access to samples of low-effort signals, which is more straightforward and practical in the LLM-influenced crowdsourcing settings. Lastly, we propose a heuristic generalization of our method to handle textual data (Appendix F), which is related to methods that elicit textual information using LLMs [Lu et al., 2024, Wu and Hartline, 2024].

The studies on LLM-content detection are also related. Common approaches for LLM content detection include watermarking [Gu et al., 2022, Lee et al., 2023, Zhao et al., 2023, Yang et al., 2023], which embeds identifiable markers into data, models, or generated text; statistical detection, which leverage differences in output by analyzing logits statistics [Su et al., 2023, Vasilatos et al., 2023] or vocabulary statistics [Gehrmann et al., 2019, Mao et al., 2024]; and deep learning-based methods, which train models to distinguish AI-generated content from human-written text [Koike et al., 2024, Hu et al., 2023, Shah et al., 2023]. However, these methods typically rely on abundant textual data for training and detection. In contrast, our approach is training-free and addresses LLM-contamination in non-textual responses scenarios, such as answering multi-choice questions.

## 3 Model

Consider a truth-discovery setting where there are $m$ i.i.d. questions. The principal (requester) distributes these questions to $n$ agents (workers) such that each agent answers multiple questions and each question is answered by more than one agent. We first present the information structure when two agents, denoted as $i$ and $j$, answer an arbitrary question. Because questions are i.i.d., the same information structure will be applied to any question.

**Signals** Suppose the underlying ground truth of the question is $Y \in \Sigma$. For example, if the answer to the question is either "yes" or "no", then $|\Sigma| = 2$. Suppose agents $i$ and $j$ both answer this question. After exerting effort, they each observe a signal $X_i$ and $X_j$ respectively. For simplicity, we assume the signal space is the same as the ground truth space, i.e. $X_i, X_j \in \Sigma$. We can interpret $X_i$ and $X_j$ as the full-effort signals that the agents can obtain which are their best guesses of $Y$.

In addition to full-effort signals, agents can obtain cheap signals $Z_i, Z_j \in \Sigma$ that may or may not depend on $Y$. Cheap signals are usually mutually correlated with each other, sometimes even more correlated than agents' full-effort signals. For example, these can be the responses of two (potentially different) LLMs.

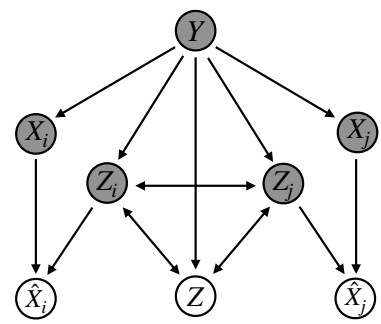

Figure 1: The causal relationship among key variables.

We present the causal relationship among these variables in Figure 1. Note that we assume $X_i$ and $Z_i$ are independent conditioned on $Y$, meaning that the full-effort signal is only correlated with the cheap signal via the ground truth. However, $Z_i$ and $Z_j$ may have additional correlations conditioned on $Y$. This may happen when agents rely on LLMs trained on similar data.

**Strategies** The principal's goal is to recover the ground truth as accurately as possible. We consider scenarios where AI underperforms human agents, motivating the principal to seek full-effort information from human agents. Such situations arise when the notion of ground truth is inherently subjective, shaped by human preferences—such as when labelers are asked to rank LLM responses according to personal preference, a common practice in reinforcement learning with human feedback.

However, agents can strategically report based on their high-effort signal $X_i$ and the cheap signal $Z_i$. Mathematically, the report $\hat{X}_i$ is a (random) function of $X_i$ and $Z_i$, denoted as $\hat{X}_i = \theta_i(X_i, Z_i)$ where $\theta_i \in \Theta$. Let $\tau$ be the truth-telling strategy, i.e. $\tau(X_i, Z_i) = X_i$, and let $\mu_i$ denote a no-effort strategy such that $\mu_i(X_i, Z_i)$ is independent of $X_i$ conditioned on $Z_i$. The argument for $\hat{X}_j$ and $\theta_j$ are analogous.

We highlight a special type of strategy, called **the *lazy-reporting strategy*, which is more practical and common on crowdsourcing platforms.**

**Definition 3.1.** A lazy-reporting strategy $\nu \in \Theta_\nu$ is a convex combination between truth-telling and a no-effort strategy $\mu$, i.e.

$$\nu(X_i, Z_i) = \begin{cases} X_i & \text{with probability } p_i < 1, \\ \mu(X_i, Z_i) & \text{otherwise.} \end{cases}$$

In particular, $\nu$ does not make $\hat{X}_i$ depend jointly on both $X_i$ and $Z_i$ on the same question. Examples of lazy-reporting strategies include: exerting effort and reporting $X_i$ for some questions while randomly selecting answers for the rest; or fully relying on an LLM and always reporting $Z_i$. In contrast, **non-lazy reporting strategies are typically complex and impractical**, often resembling malicious rather than mere low-effort behavior. For example, an agent may first exert effort to obtain $X_i$ for each question and then mix it with $Z_i$ by reporting "yes" if either $X_i$ or $Z_i$ is "yes", and "no" otherwise.

**Principal Information** The principal can observe the reports $\hat{X}_i$ and $\hat{X}_j$, but not the underlying signals $X_i, X_j$ and $Z_i, Z_j$. This implies that the principal cannot directly evaluate human responses using the ground truth $Y$. Without knowing $Y$, the principal cannot distinguish the real information $X_i$ and the cheap signal $Z_i$ without further assumptions. Our method assumes that the principal can observe a signal $Z$ that is (strongly) correlated with the agents' cheap signals. In the context of LLM contamination, for example, the principal can generate answers to a fraction of the questions using a popular LLM.

The effectiveness of our method necessarily depends on the quality of $Z$—how well the principal's information can block the correlation between agents' cheap signals. We will introduce our method and the assumptions in Section 4 and empirically verify the assumptions in Section 5.2.

### 3.1 Objectives and Problem Statement

In practice, the principal may collect responses from multiple agents, resulting in an $n \times m$ response matrix $\hat{X}$ where each entry $\hat{X}_{i,k} \in \Sigma$ if agent $i$ answers question $k$ and $\hat{X}_{i,k} = \emptyset$ otherwise. Given $\hat{X}$ and the principal's samples of the cheap signal $Z$, a scoring mechanism designs a score for each agent. **Our goal is to design a scoring mechanism that satisfies *information monotonicity*.** Let $S_i(\hat{X}_i, \hat{X}_{-i} \mid Z)$ denote the expected score of agent $i$, where $\hat{X}_{-i}$ is the reports of all agents but $i$.

**Definition 3.2.** A mechanism is *information monotone for lazy-reporting strategies* if, for any agent $i$, $S_i(X_i, X_{-i} \mid Z) > S_i(\hat{X}_i, \hat{X}_{-i} \mid Z)$ for any lazy-reporting strategy $\nu_i \in \Theta_\nu$ with $\hat{X}_i = \nu_i(X_i, Z_i)$.

A mechanism is $\epsilon$-*information monotone* if it is information monotone for lazy-reporting strategies and, additionally, $S_i(X_i, X_{-i} \mid Z) \geq S_i(\hat{X}_i, \hat{X}_{-i} \mid Z) - \epsilon$ holds for any strategy $\theta_i \in \Theta$ with $\hat{X}_i = \theta_i(X_i, Z_i)$.

At the individual level, information monotone scores can be used to identify low-effort agents. At the dataset level, the score distribution across agents can help interpret the quality of the crowdsourced dataset.

We note that the same concept is termed *informed truthfulness* or *informed strategy-proofness* in the peer prediction literature, where the objective is to incentivize agents to report truthful information Shnayder et al. [2016].

## 4 Our Method

Classic peer prediction methods often struggle with cheap signals, which are highly correlated with one another but only weakly correlated with the ground truth. In this section, we generalize a prior work—the *correlated agreement (CA) mechanism* (introduced in Appendix B)—and propose a mechanism that achieves information monotonicity when the principal can obtain some noisy samples of the cheap signal. In the context of LLM contamination, these samples can be the labels generated by a popular LLM on a fraction of the crowdsourced questions.

We call our generalization the *conditioned CA mechanism*, which scores an agent based on the correlation between her responses and her peers' conditioned on the principal's samples of $Z$. Our method has two main steps.

**Learn the scoring function.** A scoring function determines whether a pair of signals "agree" or not, which is learned based on data. We first estimate the empirical conditioned joint distribution $\tilde{P}(\hat{X}_i, \hat{X}_j \mid Z)$, computed using the frequency of observing a pair of signals on the same question conditioned on $Z$.[2] Next, we compute the delta tensor

$$\tilde{\Delta}_{h,l,k} = \tilde{P}\left(\hat{X}_i = h, \hat{X}_j = l \mid Z = k\right) - \tilde{P}\left(\hat{X}_i = h \mid Z = k\right)\tilde{P}\left(\hat{X}_j = l \mid Z = k\right),$$

which is the difference between the joint distribution and the product of marginal distributions of the reports, conditioned on $Z$. This gives us the scoring function $T_{\tilde{\Delta}} = \mathrm{sign}(\tilde{\Delta})$, i.e. $T_{\tilde{\Delta}}(h, l, k) = 1$ if $\tilde{\Delta}_{h,l,k} > 0$ and 0 otherwise. This step becomes unnecessary if the true scoring function $T_\Delta = \mathrm{sign}(\Delta)$ is already known, where $\Delta$ is defined analogously to $\tilde{\Delta}$, but based on the true signals $X_i$ and $X_j$. For example, for binary questions with positively correlated signals, $\Delta_{\cdot,\cdot,k}$ is an identity matrix for any $k \in \Sigma$.

**Compute the bonus/penalty score.** Fixing a $k \in \Sigma$, we then repeatedly sample a bonus question $q$ and a penalty question $q'$ while ensuring $Z_q = Z_{q'} = k$. For each pair of samples, the score for agent $i$ is:

$$T_{\tilde{\Delta}}(\hat{X}_{i,q}, \hat{X}_{j,q}, k) - T_{\tilde{\Delta}}(\hat{X}_{i,q}, \hat{X}_{j,q'}, k),$$

where $j$ is a randomly selected agent who answers both $q$ and $q'$. The final score for agent $i$ is obtained by averaging over all $k \in \Sigma$ and the sampled pairs of $q$ and $q'$.

Intuitively, the conditioned CA mechanism encourages agents to agree on the same question and penalizes agents when they agree on two distinct questions, given that the principal's sampled signal is identical on both questions. If agents rely purely on the cheap signal or respond independently of the true signal, the bonus score and the penalty score cancel out in expectation. This results in a score that is close to zero. We defer more details to Appendix C.

## 4.1 Theoretical Insights

We demonstrate the effectiveness of our method and clarify the conditions under which the mechanism performs reliably. Before we start, we first present an intuitive way to interpret the score computed by our mechanism. In a prior work, Kong and Schoenebeck [2019a] show that the score of the CA mechanism can be interpreted as a type of mutual information between agents' reports called the *total variation distance (TVD)* mutual information. Inspired by this prior work, we show that when agents report the high-effort signal and the underlying scoring function $T_\Delta$ is applied, the expected score computed by the conditioned CA mechanism is essentially the conditional TVD mutual information:

$$I_{\mathrm{TVD}}(X_i; X_j \mid Z) = \sum_{z \in \Sigma} \mathrm{Pr}(Z = z) \sum_{\sigma, \sigma' \in \Sigma} |\mathrm{Pr}(X_i = \sigma, X_j = \sigma' \mid z) - \mathrm{Pr}(X_i = \sigma \mid z)\mathrm{Pr}(X_j = \sigma' \mid z)|.$$

The proof of this observation can be found in Appendix D.1.

### 4.1.1 Conditions for $\epsilon$-Information Monotonicity

We first present the following pair of assumptions which require that the principal's information $Z$ can approximately block any correlation between an agent's cheap signal and the ground truth (Assumption 4.1) and the correlation between agents' cheap signals (Assumption 4.2).

**Assumption 4.1.** We assume $Z_i$ and $Y$ are approximately independent conditioned on $Z$, i.e. $|\mathrm{Pr}(Z_i, Y \mid Z) - \mathrm{Pr}(Z_i \mid Z)\mathrm{Pr}(Y \mid Z)| \leq \epsilon_i$ for any $i \in [n]$.

**Assumption 4.2.** We assume $Z_i$ and $Z_j$ are approximately independent conditioned on $Y$ and $Z$, i.e. $|\mathrm{Pr}(Z_i, Z_j \mid Y, Z) - \mathrm{Pr}(Z_i \mid Y, Z)\mathrm{Pr}(Z_j \mid Y, Z)| \leq \epsilon$ for any $i, j \in [n]$.

We first show that these two assumptions are **sufficient** to guarantee (approximate) information monotonicity.

---

[2]We can use all agents' reports to estimate the same distribution because we assume agents are homogeneous and questions are i.i.d.. This assumption is primarily made for practical concerns, as there is usually a shortage of data to estimate the distribution accurately for each pair of agents.

**Theorem 4.3.** *If $T_\Delta$ is known and $I_{TVD}(X_i; X_j \mid Z) > \hat\epsilon := \left((\epsilon + \epsilon_i + \epsilon_j)|\Sigma|^2 + (\epsilon_i + \epsilon_j)|\Sigma|^3\right)$, then under Assumption 4.1 and 4.2, the conditioned CA mechanism is $\hat\epsilon$-information monotone.*

We defer the proof to Appendix D.1. When agents report the high-effort signals truthfully, the expected score is the TVD mutual information between $X_i$ and $X_j$. Theorem 4.3 suggests that if $Z$ approximately blocks the correlation between $Z_i$ with $X_j$ and $Z_j$, then any manipulation can only decrease the TVD mutual information between $\hat X_i$ and $\hat X_j$ up to an error.

We further show the **necessity** of the assumptions for information monotonicity. The following proposition suggests that if $Z_i$ and $Z_j$ have some correlation that is not captured by $Z$, then there exists a signal structure following the relationship in Figure 1, such that the score computed by the conditioned CA mechanism is not maximized by reporting $X_i$ and $X_j$.

**Proposition 4.4.** *Suppose $T_\Delta$ is known. If there exist $(z_i, z_j, y, z)$ such that $|\Pr(Z_i = z_i, Z_j = z_j \mid Y = y, Z = z) - \Pr(Z_i = z_i \mid Y = y, Z = z)\Pr(Z_j = z_j \mid Y = y, Z = z)| = d$ for some $0 < d \le \frac{1}{4}$, then there exists a joint distribution $\Pr(X_i, X_j, Z_i, Z_j, Y, Z)$ that satisfies the causal relationship described in Figure 1 and a strategy profile $(\theta_i, \theta_j) \ne (\tau, \tau)$ such that $S_i(\hat X_i, \hat X_j \mid Z) \ge S_i(X_i, X_j \mid Z) + 8d^2$ where $\hat X_i = \theta_i(X_i, Z_i)$ and $\hat X_j = \theta_j(X_j, Z_j)$.*

Before we provide the full proof (deferred to Appendix D.2), we briefly introduce the high-level idea. Consider a signal structure with a binary signal space $\Sigma = \{0, 1\}$. Suppose agent $i$'s high-effort signal $X_i$ is perfectly accurate at predicting $X_j$ when $X_i = 0$ but is noisy when $X_i = 1$, i.e. $\Pr(X_j = 0 | X_i = 0, Z) = 1$ and $\Pr(X_j = 1 | X_i = 1, Z) < 1$. On the other hand, the cheap signal $Z_i$ is perfectly accurate at predicting $Z_j$ when $Z_i = 1$ but is noisy when $Z_i = 0$, i.e. $\Pr(Z_j = 1 | Z_i = 1, Z) = 1$ and $\Pr(Z_j = 0 | Z_i = 0, Z) < 1$. In this case, it is intuitive that agent $i$ can be better off if both agents combine their signals—reporting 0 if and only if $X_i = Z_i = 0$ and reporting 1 otherwise. Note that although this example only indicates that Assumption 4.2 is necessary, the same recipe can be straightforwardly applied to show the necessity of Assumption 4.1.

### 4.1.2 Conditions for Information Monotonicity Under Lazy-reporting Strategies

We know Assumption 4.1 and 4.2 are necessary for the conditioned CA mechanism to be $\epsilon$-information monotone. However, these assumptions are designed to guard against complex, unrealistic strategies, such as those where agents first obtain the full-effort signal and then blend it with the cheap signal. If we instead restrict our focus to lazy-reporting strategies, these assumptions on $Z$ can be significantly relaxed.

**Assumption 4.5.** For any pair of agents $i$ and $j$, we assume the conditional TVD mutual information between $X_i, X_j, Z_i, Z_j$ satisfies that

$$I_{TVD}(X_i; X_j \mid Z) > \max\left(I_{TVD}(Z_i; Z_j \mid Z), I_{TVD}(Z_i; X_j \mid Z)\right).$$

**Proposition 4.6.** *If $T_\Delta$ is known and Assumption 4.5 holds, the conditioned CA mechanism is information monotone for lazy-reporting strategies.*

We defer the proof to Appendix D.3. Intuitively, a lazy-reporting strategy is essentially a mixture of truth-telling and no-effort strategies. We show that when both agents follow lazy-reporting strategies, the score-maximizing equilibrium must fall into one of the following three cases: 1) both agents exert full effort, achieving an expected score of $I_{TVD}(X_i; X_j \mid Z)$; 2) both agents report cheap signals, with expected score $I_{TVD}(Z_i; Z_j \mid Z)$; 3) one agent exerts full effort while the other reports a cheap signal, yielding expected score $I_{TVD}(Z_i; X_j \mid Z)$. Therefore, the inequality in Assumption 4.5 ensures that full effort strictly dominates lazy-reporting strategies, establishing the proposition.

Note that Assumption 4.5 is significantly weaker than Assumption 4.1 and 4.2, which require both $I_{TVD}(Z_i; Z_j \mid Z)$ and $I_{TVD}(Z_i; X_j \mid Z)$ to be close to zero.

*Remark* 4.7. Thus far, we have assumed that the true scoring function $T_\Delta$ is known. In Appendix E, we show that the theoretical guarantees still hold even when we have to learn the scoring function from the noisy and potentially corrupted data.

## 5 Experiments

We evaluate the effectiveness of our proposed method using real-world crowdsourcing datasets. We focus on labeling tasks where human judgment remains as the gold standard, given that current LLMs

still fall short of perfect performance. We first empirically examine the conditions under which the key assumptions underlying our method are likely to hold, as well as when they may fail. Next, we simulate a mixed, noisy crowd of labelers and assess whether our method can detect unreliable labelers better than several well-established baselines. The datasets and code of our experiments are available at `https://github.com/yichiz97/LLM_contamination`.

## 5.1 Datasets

**Hatefulness/Offensiveness/Toxicity Labeling** We first consider a toxicity labeling dataset which includes $3459$ social media user comments posted in response to political news posts and videos on Twitter, YouTube, and Reddit in August 2021 [Schöpke-Gonzalez et al., 2025]. MTurk workers are assigned these comments and are requested to annotate them as hateful, offensive, and/or toxic. The definitions of hatefulness, offensiveness, and toxicity are selected from highly cited previous literature. Every question is answered by $5$ workers and each worker answers $73$ questions on average.

**Preference Alignment** We further use an alignment dataset where each question compares two LLM-generated answers to the same prompt. Labelers rate their preference on a 0–4 scale (0: A clearly better, 4: B clearly better, 2: tie). The comparison is scored from several more detailed angles, while we only focus on the overall preference. The dataset contains $10\,461$ questions, each labeled by two crowd workers and two experts. On average, each labeler answers about $180$ questions. For more details, refer to Miranda et al. [2024].

## 5.2 Assumption Verification

We empirically test the following hypotheses while varying the LLMs that generate $Z_i$, $Z_j$, and $Z$.

1. $Z_i$ and $X_j$ are approximately independent given $Z$, i.e. $I_{\text{TVD}}(Z_i; X_j \mid Z) \approx 0$.
2. $Z_i$ and $Z_j$ are approximately independent given $Z$, i.e. $I_{\text{TVD}}(Z_i; Z_j \mid Z) \approx 0$.
3. $X_i$ and $X_j$ are more correlated than $Z_i$ and $X_j$ given $Z$, i.e. $I_{\text{TVD}}(X_i; X_j \mid Z) > I_{\text{TVD}}(Z_i; X_j \mid Z)$.
4. $X_i$ and $X_j$ are more correlated than $Z_i$ and $Z_j$ given $Z$, i.e. $I_{\text{TVD}}(X_i; X_j \mid Z) > I_{\text{TVD}}(Z_i; Z_j \mid Z)$.

The first two hypotheses correspond to Assumption 4.1 and Assumption 4.2 respectively and the other two hypotheses correspond to Assumption 4.5. More broadly, our results in this section provide valuable insights into **the correlations between different LLMs and their correlations compared with human agents in the context of subjective labeling questions**.

For each considered LLM (as detailed in Appendix G.1), we prompt it to independently generate responses to each question in the dataset three times using the default temperature. These responses are treated as samples of the cheap signals $Z_i$, $Z_j$, and $Z$. Furthermore, we randomly select two human responses in the original dataset as samples of $X_i$ and $X_j$, respectively. Using these samples across all tasks, we estimate the conditioned joint distributions and then compute the conditional mutual information. We report the results for $I_{\text{TVD}}(Z_i; X_j \mid Z)$ in Figure 2, while results for $I_{\text{TVD}}(Z_i; Z_j \mid Z)$ are deferred to Appendix G.3.

**Hypotheses 1 and 2 generally do not hold.** In Figure 2, we observe that when $Z_i$ and $Z$ are sampled from different models, the mutual information between $Z_i$ and $X_j$ is often significantly higher than that expected from the random reporting (red bar). Even when $Z_i$ and $Z$ are independently sampled by the same model, conditioning on $Z$ reduces but does not eliminate this correlation. This indicates that even independently generated responses from the same LLM can remain dependent after conditioning on another independent sample, contradicting Hypothesis 1. As presented in Appendix G.3, this pattern continues to hold for $I_{\text{TVD}}(Z_i; Z_j \mid Z)$, rejecting hypothesis 2.

**Hypothesis 3 and 4 generally hold when conditioning on the "right" model.** Encouragingly, the conditioned correlation between an LLM and a human agent is consistently weaker than that between two human agents when $Z_i$ and $Z$ come from the same model. For example, in Figure 2a, the unconditional mutual information between GPT-4-generated labels and human labels exceeds that of two expert labels. However, after conditioning on GPT-4 responses, $I_{\text{TVD}}(X_i; X_j \mid Z)$ becomes significantly higher than $I_{\text{TVD}}(Z_i; X_j \mid Z)$. This pattern is consistent across all tested datasets, providing strong support for hypothesis 3. Results for Hypothesis 4 are deferred to the appendix.

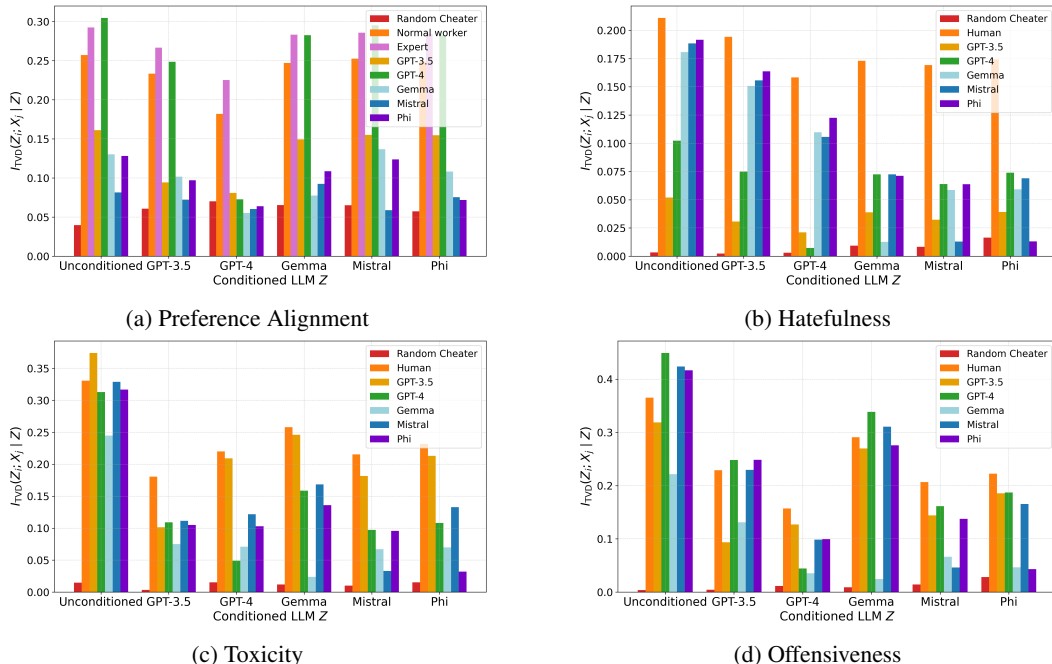

(a) Preference Alignment  (b) Hatefulness

(c) Toxicity  (d) Offensiveness

Figure 2: The TVD mutual information between $X_i$ and $X_j$ (for "Human", "Normal worker", and "expert") or $Z_i$ and $X_j$ (for remaining bars) conditioned on $Z$, i.e. $I_{\text{TVD}}(X_i; X_j \mid Z)$ or $I_{\text{TVD}}(Z_i; X_j \mid Z)$. LLM names on the x-axis denote the models used to sample $Z$ while each bar in the same group represents the model used to sample $Z_i$. We use "Random Agents" as a reference, who randomly selects an answer to each question according to the prior.

Our empirical results suggest that **as of the time of this study and based on the subjective labeling tasks we tested, human agents exhibit distinct correlation patterns that no LLM replicates.**

## 5.3 Detecting Low-effort Agents

In this subsection, we focus on identifying low-effort agents with lazy-reporting strategies and comparing various methods based on their area under the ROC curve (AUC).

### 5.3.1 Simulating Noisy Crowds

Real-world crowdsourcing datasets do not include labels identifying whether an agent is exerting low effort or working diligently. To enable evaluation, we simulate low-effort agents and treat the original dataset labels as representing "full-effort" responses. Specifically, we replace a randomly selected fraction of agents in the original dataset with simulated low-effort agents adopting lazy-reporting strategies. To reflect the noisy nature of real-world crowdsourcing data, we simulate three types of low-effort agents, each corresponding to a common lazy-reporting strategy observed in practice Yang et al. [2019], Becker et al. [2021]:

1. An $\alpha_{llm}$ fraction are **LLM-reliant agents**, who report the LLM-generated labels on all assigned questions.
2. An $\alpha_r$ fraction are **random agents**, who generate labels by independently sampling from the marginal distribution over labels for each assigned question.
3. An $\alpha_b$ fraction are **biased agents**, who report the dataset's majority label on 90% of questions and choose uniformly at random on the rest.

Figure 3a shows the distribution of conditioned CA scores, where both the LLM-reliant agents' responses and the principal's signals are independently sampled from separate GPT-4 outputs. As shown, human workers achieve higher scores than any type of low-effort agent on average. However, some workers score below the average low-effort agent. One possible explanation is that a subset of workers in the dataset may also have exerted low effort. Due to these factors, no detection method can achieve perfect accuracy.

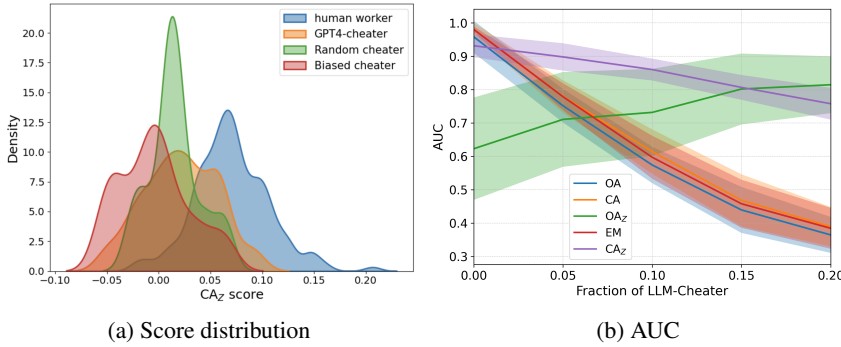

|          (a) Score distribution          |          (b) AUC          |

Figure 3: (a) Distribution of the conditioned CA scores with $(\alpha_{llm}, \alpha_r, \alpha_b) = (0.1, 0.05, 0.05)$. (b) AUC scores of various methods as the fraction of LLM-reliant agents increases. In both panels, GPT-4 generates samples for $Z_i$, $Z_j$, and $Z$ using the preference alignment dataset.

### 5.3.2 Baseline Measurements

We introduce the key ideas of the following four baselines and defer the details to Appendix G.2.

1. We consider the **Output Agreement (OA)** mechanism that directly counts the frequency of agreements between two agents' responses.
2. We use the **Correlated Agreement (CA)** mechanism as a baseline, which is introduced in Appendix B. The expected CA score is the unconditional mutual information $I_{\text{TVD}}(\hat{X}_i; \hat{X}_j)$.
3. Another common method to estimate the reliability of the noisy agents is the **Expectation-Maximization (EM)** algorithm under the DS model [Dawid and Skene, 1979]. The algorithm iteratively updates a confusion matrix that captures the noise level of each agent. After convergence, we use the trace of each agent's confusion matrix as a reliability score.
4. Lastly, we consider the **Conditioned Agreement ($\text{OA}_Z$)** which counts the frequency of agreements only when the agreement is different from the principal's sample on that question.

We note that there are many more heuristic methods to estimate the reliability of crowdsourcing agents Yin et al. [2008], Embretson and Reise [2013], Piech et al. [2013]. Many of these methods are variants of the EM algorithm under a different context-dependent model. We nonetheless only use the EM algorithm with the DS model to represent this line of methods because they all ignore the principal's information $Z$ and thus are unlikely to perform well with LLM-corrupted datasets.

### 5.3.3 Results

Given a dataset $D$ and samples of the cheap signal $Z$, we use our method as well as the baselines to score each agent. We use the AUC score to measure the ability of each discussed metric to distinguish the simulated low-effort agents from the original human agents. In particular, let $I^+$ be the set of human agents and $I^-$ be the set of simulated low-effort agents. We compute the AUC score for each method as follows:

$$\text{AUC} = \frac{1}{|I^+| \cdot |I^-|} \sum_{i \in I^+} \sum_{j \in I^-} \left( \mathbb{1}[S_i > S_j] + 0.5 \cdot \mathbb{1}[S_i = S_j] \right).$$

Figure 3b presents a comparison of different methods. For each value of $\alpha_{llm}$ ranging from $0$ to $0.2$, we randomly sample $\alpha_r$ and $\alpha_b$ uniformly from $[0, 0.2]$ and report the average and error bars over 50 such trials. Our result suggests that classic methods, such as the EM algorithm and the CA mechanism, perform well when the number of LLM-reliant agents is low. However, their AUCs degrade significantly as the fraction of LLM-reliant agents increases. On the other hand, the performance of $\text{OA}_Z$, which scores agents based on the similarity between their reports and $Z$, tends to have large variance and performs poorly when there are few LLM-reliant agents. In contrast, our proposed conditioned CA mechanism has the most robust performance for the mixed crowds and has the lowest variance.

Table 1 summarizes the results from Figure 3b, reporting the average AUC score and the bottom 10% quantile for each method on each dataset, where the average is w.r.t. every tested combination of $(\alpha_{llm}, \alpha_r, \alpha_b)$. A high average AUC reflects strong overall performance, while a high 10% quantile

indicates robustness to the worst-case scenarios. The results support our earlier claim that our method consistently demonstrates the highest robustness across all tested settings.

Table 1: Average AUC (and bottom 10% quantiles) across datasets and methods with GPT-4-reliant agents.

| Dataset | OA | CA | $OA_Z$ | EM | $CA_Z$ (ours) |
|---|---|---|---|---|---|
| Hatefulness | 0.66 (0.45) | 0.84 (0.75) | 0.75 (0.64) | 0.81 (0.70) | **0.85 (0.81)** |
| Offensiveness | 0.80 (0.68) | 0.92 (0.88) | 0.92 (0.85) | 0.89 (0.82) | **0.94 (0.91)** |
| Toxicity | 0.60 (0.40) | 0.89 (0.83) | 0.89 (0.82) | 0.85 (0.79) | **0.91 (0.87)** |
| Preference Alignment | 0.65 (0.40) | 0.70 (0.43) | 0.70 (0.45) | 0.68 (0.41) | **0.85 (0.77)** |

We note that our experiments do not claim that our method outperforms all baselines in every scenario. Traditional methods like EM often perform better when the crowd consists entirely of random or biased agents. However, our method demonstrates the most robust performance across all settings due to its theoretical guarantees. This is particularly useful as, in practice, the designer usually has little clue about the composition of the crowds. In contrast, the baselines exhibit cases where the AUC drops below 0.5, meaning the algorithm completely misclassifies good workers as low-effort agents.

**Additional results in the appendix.** First, Appendix G.4.1 presents analogous figures and tables for additional datasets and LLMs, further supporting the claims made in the main body. Second, we investigate how many questions the principal needs to sample for effective detection of low-effort agents (Appendix G.4.2). The results suggest that sampling LLM labels on 50% of the questions is typically sufficient to outperform the CA baseline. In cases where human and AI labels are well separated, sampling just 20% can suffice. Third, when the principal is uncertain about which LLMs the agents use, we propose a method that computes the CA score conditioned on each candidate LLM's outputs and uses the minimum score across them as the final score (Appendix G.4.3). Finally, in Appendix F and Appendix G.5, we present a heuristic extension of our method to open-ended textual responses. This serves as a preliminary step toward generalizing our approach beyond labeling tasks, with further refinement and large-scale validation left for future work.

## 6 Conclusion

This work addresses the challenge of LLM contamination in crowdsourcing by developing a theoretically grounded quality measure for annotation tasks. Leveraging an extension of peer prediction, our method evaluates worker responses beyond what LLMs can generate, ensuring reliable human contributions. We establish theoretical guarantees under different low-effort strategies and validate our approach empirically on real-world datasets. Our results demonstrate that the proposed method effectively detects low-effort agents while maintaining robustness across various conditions. Future work can focus on refining our method to better differentiate between harmful LLM usage and productive human-AI collaboration. This includes optimizing scoring mechanisms to recognize when LLMs are used as assistive tools rather than substitutes for human effort and developing adaptive detection techniques that adjust based on task characteristics and worker behavior.

## Broader Impact

On the positive side, our work can improve the robustness and fairness of data collection pipelines used in training AI systems, thereby enhancing the reliability of downstream decision-making or other usage of AI. However, there is a potential risk that insights into reporting strategies or detection weaknesses could be misused to evade quality control. We believe this risk is mitigated by focusing on defensive mechanisms and open discussion of vulnerabilities to promote transparency and better system design. Furthermore, like many crowdsourcing algorithms for label-quality control, our method operates without access to ground truth. Consequently, a low score does not always indicate a low-effort worker, while it may also reflect a diligent agent holding a minority opinion. Therefore, while such verification-free algorithms are highly scalable, they should be used with a human-in-the-loop, allowing the principal to carefully review low-scoring agents and prevent potential ethical concerns.

## Acknowledgment

This work was partly carried out at the DIMACS Center at Rutgers University.

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

# A    Limitations

We acknowledge several limitations of our method. First, our method is primarily designed for crowdsourcing datasets comprising subjective questions or tasks where human experts are considered the gold standard. If LLMs significantly surpass human performance, the correlations between human responses may no longer reflect the ground truth $Y$. This violates the assumption that, conditioned on $Y$, agents' high-effort signals are independent. In such cases, applying our method could mistakenly reward incorrect correlations.

Second, as discussed in our experiment section, our method does not outperform all baselines in every scenario. In particular, when LLM responses are diverse, e.g. when each agent relies on a different model or only a few agents use LLM assistance, heuristic methods such as the EM algorithm work well. This is because diverse or sparse low-effort agents are less correlated with the majority of the crowd and can be easily flagged as outliers using the classic methods. Despite this, our method demonstrates the most robust performance across a wide range of settings. Future work could incorporate the principal's sampled LLM signals into heuristic methods such as EM to improve their performance and enhance detection under LLM contamination.

Third, our experiments evaluate only five language models, as some models, such as LLaMA and Gemini, refuse to respond to toxicity-related prompts. While this is sufficient to demonstrate the method's effectiveness, additional evaluations across more diverse models and task types would provide a more comprehensive assessment.

# B    Preliminary

What do we know about the problem? Our work builds on advancements in information elicitation, particularly peer prediction. First introduced by Miller et al. [2005], peer prediction mechanisms aim to incentivize truthful reporting without requiring ground truth verification. The core idea is to reward agents based on how well their responses predict those of others. The original mechanism relied on the principal's knowledge of the correlation between agents' signals (the information they observe). Later, Dasgupta and Ghosh [2013] extended this idea, removing the need for such knowledge by assuming agents answer multiple i.i.d. binary questions. Subsequent works have expanded this framework to non-binary (but finite) signals [Shnayder et al., 2016, Kong and Schoenebeck, 2019b], infinite signals [Schoenebeck and Yu, 2020], truthful guarantees with fewer questions [Kong, 2020b], heterogeneous agents [Agarwal et al., 2017], adversarial agents [Schoenebeck et al., 2021], and more general reporting strategies [Zhang and Schoenebeck, 2023b].

Under the assumption that there are no cheap signals, the peer prediction literature suggests that estimating the mutual information between $X_i$ and $X_j$ is an information monotone metric Kong and Schoenebeck [2019a]. There are various ways to estimate the mutual information using agents' responses to multiple questions Kong [2020a], Schoenebeck and Yu [2020] and many information monotone metrics that can be explained using mutual information Dasgupta and Ghosh [2013], Shnayder et al. [2016]. In this section, we discuss a simple yet intuitive mechanism called the correlated agreement (CA) mechanism [Shnayder et al., 2016].

The CA mechanism rewards agents when they "agree" on the same question and penalizes them when they "agree" on distinct questions. Instead of directly checking agreement, the mechanism redefines agreement based on the correlation between signals. A pair of signals $\sigma, \sigma' \in \Sigma$ is correlated agreed if $\Pr(X_i = \sigma, X_j = \sigma') - \Pr(X_i = \sigma)\Pr(X_j = \sigma') > 0$, meaning that $\sigma$ and $\sigma'$ are more likely to appear on the same question rather than on distinct questions. In particular, the mechanism utilizes the delta matrix $\Delta_{\sigma,\sigma'} = \Pr(X_i = \sigma, X_j = \sigma') - \Pr(X_i = \sigma)\Pr(X_j = \sigma')$ to design the scoring function $T_\Delta = \text{sign}(\Delta)$ where $\text{sign}(x) = 1$ if $x > 0$ and 0 otherwise.

In practice, the mechanism can be implemented in two ways depending on the information available to the principal. First, if $T_\Delta$ is known, the mechanism can be directly implemented, as shown in Algorithm 1. This case happens if the information structure is simple, e.g. when questions have binary answers, $T_\Delta$ is likely to be a $2 \times 2$ identical matrix. Second, if $T_\Delta$ is unknown, the mechanism can first estimate the joint distribution, denoted as $\widetilde{\Pr}(\hat{X}_i, \hat{X}_j)$ using the report matrix $\hat{X}$. Then, an estimate of $T_\Delta$ can be computed using the estimated Delta matrix $\tilde{\Delta}_{\sigma,\sigma'} = \widetilde{\Pr}(\hat{X}_i = \sigma, \hat{X}_j = \sigma') - \widetilde{\Pr}(\hat{X}_i = \sigma)\widetilde{\Pr}(\hat{X}_j = \sigma')$. Next, Algorithm 1 can be applied.

We emphasize that penalizing agreements on distinct questions is crucial for the CA mechanism to maintain information monotonicity. Without such penalization, over-reporting the majority signal could result in a high score. This form of manipulation is particularly problematic when the dataset is biased. For instance, in a task to label messages as toxic or not, if only 5% of the data points are toxic, consistently reporting "not toxic" could yield a higher score than truthfully reporting the actual signal.

Therefore, the score computed by the CA mechanism can be interpreted in the following way. Let the marginal distribution of agent $i$'s reports be $\Pr(\hat{X}_i)$. Imagine an agent $k$ who reports randomly for each question according to $\Pr(\hat{X}_i)$. Then, the CA score for agent $i$ can be understood as the average number of questions on which agent $i$ agrees with another randomly selected agent $j$ more than agent $k$ agrees with $j$.

---

**Algorithm 1:** The Correlated Agreement Mechanism [Shnayder et al., 2016]

---

**Input:** The crowdsourced dataset $\hat{X}$, the (estimated) scoring function $T_{\tilde{\Delta}}$
**Output:** A score vector $s$
**for** *agent* $i \in [n]$ **do**
    Let $M_i$ be the set of questions answered by agent $i$
    Initialize $x \leftarrow 0$
    **for** *question* $q \in M_i$ **do**
        Randomly select a peer $j$ who answered question $q$
        $x \leftarrow x + T_{\tilde{\Delta}}(\hat{X}_{i,q}, \hat{X}_{j,q})$ Randomly select a question $q' \neq q$ answered by agent $j$
        $x \leftarrow x - T_{\tilde{\Delta}}(\hat{X}_{i,q}, \hat{X}_{j,q'})$
    $s_i \leftarrow x/|M_i|$
**return** $s$

---

## C   Details of the Conditioned Correlated Agreement Mechanism

After learning the scoring function, the conditioned CA mechanism scores agents according to Algorithm 2. At a high level, the mechanism will iteratively fix a signal $k \in \Sigma$ and look at the questions on which $Z_q = k$. Let $M_Z(k)$ be the set of questions on which $Z = k$. Then, an agent $i$ will receive a higher score if she agrees with another agent $j$ on the same question more often than on two distinct questions conditioning that the questions are drawn from $M_Z(k)$. This measures the additional correlation between two independent agents conditioned on $Z$.

---

**Algorithm 2:** The conditioned CA mechanism

---

**Input:** The crowdsourced dataset $\hat{X}$, the samples of cheap signal $Z$, the (estimated) scoring function $T_{\tilde{\Delta}}$
Estimate the empirical marginal distribution of $Z$, i.e. $\tilde{P}(Z = \sigma)$ for $\sigma \in \Sigma$.
**for** *agent* $i \in [n]$ **do**
    Let $M_i$ be the set of questions answered by agent $i$;
    Initialize $s_i = 0$;
    **for** $k \in \Sigma$ **do**
        If $M_i \cap M_Z(k) = \emptyset$, **continue**;
        Initialize $x = 0$;
        **for** *question* $q \in M_i \cap M_Z(k)$ **do**
            Randomly select a peer $j$ who answers $q$;
            $x \mathrel{+}= T_{\tilde{\Delta}}(\hat{X}_{i,q}, \hat{X}_{j,q}, k)$;
            Randomly select a question $q' \in M_j \cap M_Z(k)$;
            $x \mathrel{-}= T_{\tilde{\Delta}}(\hat{X}_{i,q}, \hat{X}_{j,q'}, k)$
        $s_i \mathrel{+}= x \cdot \tilde{P}(Z = k)/|M_i \cap M_Z(k)|$
**return** $s = (s_1, \ldots, s_n)$

---

## D  Proofs

### D.1  Sufficiency of the Assumptions

*Proof of Theorem 4.3.* Intuitively, Assumption 4.1 and Assumption 4.2 imply that conditioned on $Z$, $Z_i$ is almost independent of $X_i$, $X_j$, and $Z_j$. Therefore, any manipulation that makes $\hat{X}_i$ depend on $Z_i$ will not increase the expected score. To prove this, we first write down the expected score when agent $i$ and $j$ play strategy $\theta_i$ and $\theta_j$ respectively.

$$
\begin{aligned}
&S_i(\theta_i, \theta_j) \\
=& \sum_{z \in \Sigma} \Pr(Z = z) \sum_{x_i, x_j, z_i, z_j \in \Sigma} \underbrace{\Pr(X_i = x_i, X_j = x_j, Z_i = z_i, Z_j = z_j \mid z) \, T_\Delta(\theta_i(x_i, z_i), \theta_j(x_j, z_j), z)}_{\text{the expected score of the bonus question}} \\
&- \sum_{z \in \Sigma} \Pr(Z = z) \sum_{x_i, x_j, z_i, z_j \in \Sigma} \underbrace{\Pr(X_i = x_i, Z_i = z_i \mid z) \, \Pr(X_j = x_j, Z_j = z_j \mid z) \, T_\Delta(\theta_i(x_i, z_i), \theta_j(x_j, z_j), z)}_{\text{the expected score of the penalty question}}.
\end{aligned}
$$

For comparison, we next write down the expected score when both agents report truthfully.

$$
\begin{aligned}
S_i(\tau, \tau) =& \sum_{z \in \Sigma} \Pr(Z = z) \sum_{x_i, x_j \in \Sigma} \underbrace{\Pr(X_i = x_i, X_j = x_j \mid z) \, T_\Delta(x_i, x_j, z)}_{\text{the expected score of the bonus question}} \\
&- \sum_{z \in \Sigma} \Pr(Z = z) \sum_{x_i, x_j \in \Sigma} \underbrace{\Pr(X_i = x_i \mid z) \, \Pr(X_j = x_j \mid z) \, T_\Delta(x_i, x_j, z)}_{\text{the expected score of the penalty question}} \\
=& \sum_{z \in \Sigma} \Pr(Z = z) \sum_{x_i, x_j \in \Sigma} (\Pr(x_i, x_j \mid z) - \Pr(x_i \mid z) \, \Pr(x_j \mid z)) \, T_\Delta(x_i, x_j, z) \\
=& \sum_{z \in \Sigma} \Pr(Z = z) \sum_{x_i, x_j \in \Sigma} |\Pr(x_i, x_j \mid z) - \Pr(x_i \mid z) \, \Pr(x_j \mid z)| \\
=& I_{\text{TVD}}(X_i; X_j \mid Z)
\end{aligned}
$$

We want to show that $S_i(\theta_i, \theta_j) \leq S_i(\tau, \tau)$ up to an error. As shown above, the expected score can be decomposed into the expected score of the bonus question and the expected score of the penalty question. Next, we provide an upper bound on the bonus score and a lower bound on the penalty score.

**Lower bound the penalty score.** We first provide a lower bound for $\Pr(X_i, Z_i \mid Z)$, which is summarized in the following lemma.

**Lemma D.1.** *Under Assumption 4.1,* $\Pr(X_i, Z_i \mid Z) \geq \Pr(Z_i \mid Z) \Pr(X_i \mid Z) - \epsilon_i \sum_{y \in \Sigma} \Pr(X_i \mid Y = y, Z)$.

Then, we provide a lower bound of the penalty score while fixing a $Z = z$.

$$
\sum_{x_i, x_j, z_i, z_j \in \Sigma} \Pr(x_i, z_i \mid z) \, \Pr(x_j, z_j \mid z) \, T_\Delta(\theta_i(x_i, z_i), \theta_j(x_j, z_j), z)
$$

$$
\geq \sum_{x_i, x_j, z_i, z_j \in \Sigma} \left( \Pr(z_i \mid z) \Pr(x_i \mid z) - \epsilon_i \sum_{y \in \Sigma} \Pr(x_i \mid y, z) \right) \Pr(x_j, z_j \mid z) \, T_\Delta(\theta_i(x_i, z_i), \theta_j(x_j, z_j), z).
$$

$$
\text{(By Lemma D.1)}
$$

By reordering the summations,

$$
= \sum_{x_i, x_j, z_i, z_j \in \Sigma} \Pr(z_i \mid z) \Pr(x_i \mid z) \, \Pr(x_j, z_j \mid z) \, T_\Delta(\theta_i(x_i, z_i), \theta_j(x_j, z_j), z)
$$

$$
- \epsilon_i \sum_{y, z_i \in \Sigma} \sum_{x_j, z_j \in \Sigma} \Pr(x_j, z_j \mid z) \sum_{x_i \in \Sigma} \Pr(x_i \mid y, z) T_\Delta(\theta_i(x_i, z_i), \theta_j(x_j, z_j), z).
$$

We focus on the second term. First note that $T_\Delta \leq 1$. Therefore, the second term is lower bounded by $-\epsilon_i \sum_{y,z_i \in \Sigma} \sum_{x_j,z_j \in \Sigma} \Pr(x_j, z_j \mid z) \sum_{x_i \in \Sigma} \Pr(x_i \mid y, z)$. Marginalizing over $x_j, z_j$ and $x_i$ and observing that $y, z_i \in \Sigma$, the quality is lower-bounded by $\epsilon_i |\Sigma|^2$. This means the original quantity is lower-bounded by

$$\geq \sum_{x_i,x_j,z_i,z_j \in \Sigma} \Pr(z_i \mid z) \Pr(x_i \mid z)\ \Pr(x_j, z_j \mid z)\ T_\Delta(\theta_i(x_i, z_i), \theta_j(x_j, z_j), z) - \epsilon_i |\Sigma|^2.$$

Next, we apply the same derivation for $j$.

$$= \sum_{x_i,x_j,z_i,z_j \in \Sigma} \Pr(z_i \mid z) \Pr(x_i \mid z) \left( \Pr(z_j \mid z) \Pr(x_j \mid z) - \epsilon_j \sum_{y \in \Sigma} \Pr(x_j \mid y, z) \right) \tag{1}$$

$$\cdot T_\Delta(\theta_i(x_i, z_i), \theta_j(x_j, z_j), z) - \epsilon_i |\Sigma|^2 \qquad \text{(By Lemma D.1)}$$

$$\geq \sum_{x_i,x_j,z_i,z_j \in \Sigma} \Pr(z_i \mid z) \Pr(x_i \mid z) \Pr(z_j \mid z) \Pr(x_j \mid z)\ T_\Delta(\theta_i(x_i, z_i), \theta_j(x_j, z_j), z) - (\epsilon_i + \epsilon_j)|\Sigma|^2.$$

$$\tag{2}$$

**Upper bound the bonus score.** We first present a lemma that bounds the difference between $\Pr(Z_i, Z_j \mid Y, Z)$ and $\Pr(Z_i \mid Z) \Pr(Z_j \mid Z)$.

**Lemma D.2.** *Under Assumption 4.1 and Assumption 4.2,* $|\Pr(Z_i, Z_j \mid Y, Z) - \Pr(Z_i \mid Z) \Pr(Z_j \mid Z)| \leq \epsilon + \frac{\epsilon_i + \epsilon_j}{\Pr(Y|Z)}$.

With this lemma, we further provide an upper bound of the expected score on the bonus question. Fixing $Z = z$,

$$\sum_{x_i,x_j,z_i,z_j \in \Sigma} \Pr(x_i, x_j, z_i, z_j \mid z)\ T_\Delta(\theta_i(x_i, z_i), \theta_j(x_j, z_j), z)$$

$$= \sum_{x_i,x_j,z_i,z_j \in \Sigma} \sum_{y \in \Sigma} \Pr(y \mid z) \Pr(x_i, x_j, z_i, z_j \mid y, z)\ T_\Delta(\theta_i(x_i, z_i), \theta_j(x_j, z_j), z)$$

$$\text{(By the law of total probability)}$$

$$= \sum_{x_i,x_j,z_i,z_j \in \Sigma} \sum_{y \in \Sigma} \Pr(y \mid z) \Pr(x_i, x_j \mid y, z) \Pr(z_i, z_j \mid y, z)\ T_\Delta(\theta_i(x_i, z_i), \theta_j(x_j, z_j), z)$$

$$\text{($X$ and $Z$ are independent conditioned on $Y$.)}$$

$$\leq \sum_{x_i,x_j,z_i,z_j \in \Sigma} \sum_{y \in \Sigma} \Pr(y \mid z) \Pr(x_i, x_j \mid y, z) \left( \Pr(z_i \mid z) \Pr(z_j \mid z) + \epsilon + \frac{\epsilon_i + \epsilon_j}{\Pr(y|z)} \right) T_\Delta(\theta_i(x_i, z_i), \theta_j(x_j, z_j), z).$$

$$\text{(By Lemma D.2)}$$

We can further write this into three terms.

$$= \sum_{x_i,x_j,z_i,z_j \in \Sigma} \sum_{y \in \Sigma} \Pr(y \mid z) \Pr(x_i, x_j \mid y, z) \Pr(z_i \mid z) \Pr(z_j \mid z)\ T_\Delta(\theta_i(x_i, z_i), \theta_j(x_j, z_j), z)$$

$$\tag{3}$$

$$+ \epsilon \sum_{x_i,x_j,z_i,z_j \in \Sigma} \sum_{y \in \Sigma} \Pr(y \mid z) \Pr(x_i, x_j \mid y, z)\ T_\Delta(\theta_i(x_i, z_i), \theta_j(x_j, z_j), z) \tag{4}$$

$$+ (\epsilon_i + \epsilon_j) \sum_{x_i,x_j,z_i,z_j \in \Sigma} \sum_{y \in \Sigma} \Pr(x_i, x_j \mid y, z)\ T_\Delta(\theta_i(x_i, z_i), \theta_j(x_j, z_j), z). \tag{5}$$

First note that

$$Equation\ (3) = \sum_{x_i,x_j,z_i,z_j \in \Sigma} \Pr(z_i \mid z) \Pr(z_j \mid z)\ T_\Delta(\theta_i(x_i, z_i), \theta_j(x_j, z_j), z) \sum_{y \in \Sigma} \Pr(x_i, x_j, y \mid z)$$

$$= \sum_{x_i,x_j,z_i,z_j \in \Sigma} \Pr(z_i \mid z) \Pr(z_j \mid z)\ T_\Delta(\theta_i(x_i, z_i), \theta_j(x_j, z_j), z) \Pr(x_i, x_j \mid z)$$

Next, we upper bound the second and the third term respectively. By reordering the summations,

$$Equation~(4) = \epsilon \sum_{x_i, x_j, z_i, z_j \in \Sigma} T_\Delta(\theta_i(x_i, z_i), \theta_j(x_j, z_j), z) \sum_{y \in \Sigma} \Pr(x_i, x_j, y \mid z)$$

$$= \epsilon \sum_{x_i, x_j, z_i, z_j \in \Sigma} T_\Delta(\theta_i(x_i, z_i), \theta_j(x_j, z_j), z) \; \Pr(x_i, x_j \mid z)$$

$$\leq \epsilon \sum_{x_i, x_j, z_i, z_j \in \Sigma} \Pr(x_i, x_j \mid z) \qquad\qquad (T_\Delta \leq 1)$$

$$\leq \epsilon |\Sigma|^2. \qquad\qquad (\textstyle\sum_{x_i, x_j \in \Sigma} \Pr(x_i, x_j \mid z) = 1)$$

$$Equation~(5) = (\epsilon_i + \epsilon_j) \sum_{y \in \Sigma} \sum_{z_i, z_j \in \Sigma} \sum_{x_i, x_j \in \Sigma} \Pr(x_i, x_j \mid y, z) T_\Delta(\theta_i(x_i, z_i), \theta_j(x_j, z_j), z)$$

$$\leq (\epsilon_i + \epsilon_j) \sum_{y \in \Sigma} \sum_{z_i, z_j \in \Sigma} \sum_{x_i, x_j \in \Sigma} \Pr(x_i, x_j \mid y, z) \qquad\qquad (T_\Delta \leq 1)$$

$$\leq (\epsilon_i + \epsilon_j) |\Sigma|^3. \qquad\qquad (\textstyle\sum_{x_i, x_j \in \Sigma} \Pr(x_i, x_j \mid y, z) = 1)$$

Combining everything,

$$\sum_{x_i, x_j, z_i, z_j \in \Sigma} \Pr(x_i, x_j, z_i, z_j \mid z) \; T_\Delta(\theta_i(x_i, z_i), \theta_j(x_j, z_j), z)$$

$$\leq \sum_{x_i, x_j, z_i, z_j \in \Sigma} \Pr(z_i \mid z) \Pr(z_j \mid z) \; \Pr(x_i, x_j \mid z) \; T_\Delta(\theta_i(x_i, z_i), \theta_j(x_j, z_j), z) + \epsilon |\Sigma|^2 + (\epsilon_i + \epsilon_j) |\Sigma|^3.$$

$$(6)$$

Finally, combining Equation (2) and Equation (6),

$$S_i(\theta_i, \theta_j)$$

$$\leq \sum_{z \in \Sigma} \Pr(Z = z) \sum_{z_i, z_j \in \Sigma} \Pr(z_i \mid z) \Pr(z_j \mid z) \sum_{x_i, x_j \in \Sigma} (\Pr(x_i, x_j \mid z) - \Pr(x_i \mid z) \Pr(x_j \mid z))$$

$$\cdot T_\Delta(\theta_i(x_i, z_i), \theta_j(x_j, z_j), z) + (\epsilon + \epsilon_i + \epsilon_j) |\Sigma|^2 + (\epsilon_i + \epsilon_j) |\Sigma|^3 \qquad\qquad (7)$$

For simplicity, let $\hat{\epsilon} = (\epsilon + \epsilon_i + \epsilon_j) |\Sigma|^2 + (\epsilon_i + \epsilon_j) |\Sigma|^3$.

$$= \sum_{z \in \Sigma} \Pr(Z = z) \sum_{z_i, z_j \in \Sigma} \Pr(z_i \mid z) \Pr(z_j \mid z) \sum_{x_i, x_j \in \Sigma} \Delta_{x_i, x_j, z} \; T_\Delta(\theta_i(x_i, z_i), \theta_j(x_j, z_j), z) + \hat{\epsilon}$$

$$\leq \sum_{z \in \Sigma} \Pr(Z = z) \sum_{z_i, z_j \in \Sigma} \Pr(z_i \mid z) \Pr(z_j \mid z) \sum_{x_i, x_j \in \Sigma} \Delta_{x_i, x_j, z} \; T_\Delta(x_i, x_j, z) + \hat{\epsilon}$$

$$\text{(By the design of } T_\Delta)$$

$$= \sum_{z \in \Sigma} \Pr(Z = z) \sum_{x_i, x_j \in \Sigma} \Delta_{x_i, x_j, z} \; T_\Delta(x_i, x_j, z) + \hat{\epsilon}$$

$$= S_i(\tau, \tau) + \hat{\epsilon}.$$

Lastly, we show that if agent $i$ plays a lazy-reporting strategy $\nu_i$, the expected score is strictly smaller than truth-telling. Note that because a lazy-reporting strategy is a convex combination between truth-telling $\tau$ and a no-effort strategy $\mu_i$, it suffices to show that the expected score of any no-effort strategy is strictly smaller than truth-telling. Because $\mu_i(X_i, Z_i)$ is independent of $X_i$, we can reorder

the summations in Equation (7) in the following way. Let $\mu = \mu_i(X_i, Z_i)$.

$$S_i(\mu_i, \theta_j)$$

$$\leq \sum_{z \in \Sigma} \Pr(Z = z) \sum_{z_i, z_j \in \Sigma} \Pr(z_i \mid z) \Pr(z_j \mid z) \sum_{x_j \in \Sigma} T_\Delta(\mu, \theta_j(x_j, z_j), z)$$

$$\sum_{x_i \in \Sigma} (\Pr(x_i, x_j \mid z) - \Pr(x_i \mid z) \Pr(x_j \mid z)) + \hat{\epsilon} \qquad \text{(by Equation (7))}$$

$$= \sum_{z \in \Sigma} \Pr(Z = z) \sum_{z_i, z_j \in \Sigma} \Pr(z_i \mid z) \Pr(z_j \mid z) \sum_{x_j \in \Sigma} T_\Delta(\mu, \theta_j(x_j, z_j), z) (\Pr(x_j \mid z) - \Pr(x_j \mid z)) + \hat{\epsilon}$$

$$= \hat{\epsilon}.$$

Note that the expected score under truth-telling is exactly $I_{\text{TVD}}(X_i; X_j \mid Z)$. Therefore, whenever $I_{\text{TVD}}(X_i; X_j \mid Z) > \hat{\epsilon}$, we have $S_i(\tau, \tau) > S_i(\mu_i, \theta_j)$ for any no-effort strategy $\mu_i$ and any strategy $\theta_j$. This completes the proof. $\qquad\square$

*Proof of Lemma D.1.*

$$\Pr(X_i, Z_i \mid Z) = \sum_{y \in \Sigma} \Pr(Y = y \mid Z) \Pr(X_i, Z_i \mid Y = y, Z) \qquad \text{(By the law of total probability)}$$

$$= \sum_{y \in \Sigma} \Pr(Y = y \mid Z) \Pr(X_i \mid Y = y, Z) \Pr(Z_i \mid Y = y, Z)$$

$$(X_i \text{ and } Z_i \text{ are independent conditioned on } Y.)$$

$$\geq \sum_{y \in \Sigma} \Pr(Y = y \mid Z) \Pr(X_i \mid Y = y, Z) \frac{\Pr(Z_i \mid Z) \Pr(Y = y \mid Z) - \epsilon_i}{\Pr(Y = y \mid Z)}$$

$$\text{(By Assumption 4.1)}$$

$$= \Pr(Z_i \mid Z) \sum_{y \in \Sigma} \Pr(Y = y \mid Z) \Pr(X_i \mid Y = y, Z) - \epsilon_i \sum_{y \in \Sigma} \Pr(X_i \mid Y = y, Z)$$

$$= \Pr(Z_i \mid Z) \Pr(X_i \mid Z) - \epsilon_i \sum_{y \in \Sigma} \Pr(X_i \mid Y = y, Z).$$

$$\square$$

*Proof of Lemma D.2.*

$$|\Pr(Z_i, Z_j \mid Y, Z) - \Pr(Z_i \mid Z) \Pr(Z_j \mid Z)|$$
$$= |\Pr(Z_i, Z_j \mid Y, Z) - \Pr(Z_i \mid Y, Z) \Pr(Z_j \mid Y, Z) + \Pr(Z_i \mid Y, Z) \Pr(Z_j \mid Y, Z) - \Pr(Z_i \mid Z) \Pr(Z_j \mid Z)|$$
$$\leq |\Pr(Z_i, Z_j \mid Y, Z) - \Pr(Z_i \mid Y, Z) \Pr(Z_j \mid Y, Z)| + |\Pr(Z_i \mid Y, Z) \Pr(Z_j \mid Y, Z) - \Pr(Z_i \mid Z) \Pr(Z_j \mid Z)|$$
$$\text{(Triangle inequality)}$$
$$\leq \epsilon + |\Pr(Z_i \mid Y, Z) \Pr(Z_j \mid Y, Z) - \Pr(Z_i \mid Z) \Pr(Z_j \mid Z)|. \qquad \text{(Assumption 4.2)}$$

Therefore, we only have to bound the second quantity.

$$|\Pr(Z_i \mid Y, Z) \Pr(Z_j \mid Y, Z) - \Pr(Z_i \mid Z) \Pr(Z_j \mid Z)|$$
$$= |\Pr(Z_i \mid Y, Z)(\Pr(Z_j \mid Y, Z) - \Pr(Z_j \mid Z)) + \Pr(Z_j \mid Z)(\Pr(Z_i \mid Y, Z) - \Pr(Z_i \mid Z))|$$
$$\leq |\Pr(Z_i \mid Y, Z)(\Pr(Z_j \mid Y, Z) - \Pr(Z_j \mid Z))| + |\Pr(Z_j \mid Z)(\Pr(Z_i \mid Y, Z) - \Pr(Z_i \mid Z))|$$
$$\text{(Triangle inequality)}$$
$$\leq |\Pr(Z_j \mid Y, Z) - \Pr(Z_j \mid Z)| + |\Pr(Z_i \mid Y, Z) - \Pr(Z_i \mid Z)|$$
$$= \frac{1}{\Pr(Y \mid Z)} |\Pr(Z_j, Y \mid Z) - \Pr(Y \mid Z) \Pr(Z_j \mid Z)| + \frac{1}{\Pr(Y \mid Z)} |\Pr(Z_i, Y \mid Z) - \Pr(Y \mid Z) \Pr(Z_i \mid Z)|$$
$$\leq \frac{\epsilon_1 + \epsilon_2}{\Pr(Y \mid Z)}. \qquad \text{(By Assumption 4.1)}$$

Combining everything, we complete the proof. $\qquad\square$

## D.2 Necessity of the Assumptions

*Proof of Proposition 4.4.* Intuitively, if $Z_i$ and $Z_j$ are correlated in a different way from $X_i$ and $X_j$, then it makes $Z_i$ and $Z_j$ no longer "cheap" signals. In this case, a strategy that combines $Z_i$ and $X_i$ will improve the expected score.

Suppose $\Sigma = \{0, 1\}$ contains two signals. Consider the following joint distribution for $\Pr((X_i, Z_i), (X_j, Z_j)|Y = y, Z = z)$ where row 1 to 4 corresponds to $(X_i, Z_i) = (0, 0), (0, 1), (1, 0), (1, 1)$ respectively and the columns stand for $(X_j, Z_j)$. For simplicity, let's suppose the following joint distribution is identical for any $y$ and $z$.

$$\Pr((X_i, Z_i), (X_j, Z_j) \mid y, z) = \begin{pmatrix} 2d & 0 & 0 & 0 \\ 0 & 0 & 0 & 0 \\ 0 & 0 & \frac{1}{2} - 2d & \frac{1}{2} - 2d \\ 0 & d & 0 & d \end{pmatrix} \quad \text{for any } y, z \in \{0, 1\}.$$

Note that the example is valid when $0 < d \leq \frac{1}{4}$. Using this joint distribution, we can further write down the joint distribution between other pairs of variables.

$$\Pr(X_i, X_j \mid y, z) = \begin{pmatrix} 2d & 0 \\ d & 1 - 3d \end{pmatrix} \qquad \Pr(Z_i, Z_j \mid y, z) = \begin{pmatrix} \frac{1}{2} & \frac{1}{2} - 2d \\ 0 & 2d \end{pmatrix} \quad \text{for any } y, z \in \{0, 1\}.$$

Furthermore, the product of the marginal distributions are

$$\Pr(X_i \mid y, z) \Pr(X_j \mid y, z) = \begin{pmatrix} 6d^2 & 2d(1 - 3d) \\ 3d(1 - 2d) & (1 - 2d)(1 - 3d) \end{pmatrix}$$

$$\Pr(Z_i \mid y, z) \Pr(Z_j \mid y, z) = \begin{pmatrix} \frac{1}{2}(1 - 2d) & \frac{1}{2}(1 - 2d) \\ d & d \end{pmatrix} \quad \text{for any } y, z \in \{0, 1\}.$$

We can further write down the difference between the joint distribution and the product of the marginal distributions.

$$\Pr(X_i, X_j \mid y, z) - \Pr(X_i \mid y, z) \Pr(X_j \mid y, z) = \begin{pmatrix} 2d(1 - 3d) & -2d(1 - 3d) \\ -2d(1 - 3d) & 2d(1 - 3d) \end{pmatrix}$$

$$\Pr(Z_i, Z_j \mid y, z) - \Pr(Z_i \mid y, z) \Pr(Z_j \mid y, z) = \begin{pmatrix} d & -d \\ -d & d \end{pmatrix}$$

Note that in this example, $|\Pr(Z_i, Z_j \mid y, z) - \Pr(Z_i \mid y, z) \Pr(Z_j \mid y, z)| = d$ for any $Z_i, Z_j \in \{0, 1\}$, satisfying the condition in the proposition. Furthermore, for any $k \in \{0, 1\}$, the scoring function $T_\Delta(k, h, l) = 1$ if and only if $h = l$ and 0 otherwise. Now, we compute the expected score when both agents honestly report the high-effort signal.

$$
\begin{aligned}
S_i(X_i, X_j \mid Z) &= \sum_{z \in \{0,1\}} \Pr(Z = z) \\
&\qquad \sum_{x_i, x_j \in \{0,1\}} (\Pr(X_i = x_i, X_j = x_j \mid z) - \Pr(X_i = x_i \mid z) \Pr(X_j = x_j \mid z)) T_\Delta(x_i, x_j, z) \\
&= \sum_{x_i, x_j \in \{0,1\}} |\Pr(x_i, x_j \mid z) - \Pr(x_i \mid z) \Pr(x_j \mid z)| \\
&= 8d(1 - 3d).
\end{aligned}
$$

We show that this score is strictly smaller than the expected score when agents report 0 if and only if $X_i = Z_i = 0$ (and $X_j = Z_j = 0$) and report 1 otherwise. Let $\theta$ be the strategy such that $\theta(k, l|z) = 0$ if and only if $k = l = 0$ for any $z$. We can write down the joint distribution and the product of marginal distributions for $\hat{X}_i$ and $\hat{X}_j$.

$$\Pr(\hat{X}_i, \hat{X}_j \mid y, z) = \begin{pmatrix} 2d & 0 \\ 0 & 1 - 2d \end{pmatrix}$$

$$\Pr(\hat{X}_i \mid y, z) \Pr(\hat{X}_j \mid y, z) = \begin{pmatrix} 4d^2 & 2d(1 - 2d) \\ 2d(1 - 2d) & (1 - 2d)^2 \end{pmatrix} \quad \text{for any } y, z \in \{0, 1\}.$$

The difference between the above two matrices is thus

$$\Pr(\hat{X}_i, \hat{X}_j \mid y, z) - \Pr(\hat{X}_i \mid y, z)\Pr(\hat{X}_j \mid y, z) = \begin{pmatrix} 2d(1-2d) & -2d(1-2d) \\ -2d(1-2d) & 2d(1-2d) \end{pmatrix} \quad \text{for any } y, z \in \{0,1\}.$$

This means that the expected score under the strategy profile $(\theta, \theta)$ is

$$
\begin{aligned}
S_i(\hat{X}_i, \hat{X}_j \mid Z) &= \sum_{z \in \{0,1\}} \Pr(Z = z) \\
&\qquad \sum_{x_i, x_j \in \{0,1\}} \left( \Pr((\hat{X}_i = x_i, \hat{X}_j = x_j \mid z) - \Pr(\hat{X}_i = x_i \mid z)\,\Pr(\hat{X}_j = x_j \mid z) \right) T_\Delta(x_i, x_j, z) \\
&= \sum_{x_i, x_j \in \{0,1\}} \left| \Pr(\hat{X}_i = x_i, \hat{X}_j = x_j \mid z) - \Pr(\hat{X}_i = x_i \mid z)\,\Pr(\hat{X}_j = x_j \mid z) \right| \\
&= 8d(1-2d) \\
&= S_i(X_i, X_j \mid Z) + 8d^2.
\end{aligned}
$$

$\square$

### D.3 Information Monotonicity under Lazy-reporting Strategies

*Proof of Proposition 4.6.* A lazy-reporting strategy is a convex combination of truth-telling and a no-effort strategy. Therefore, it suffices to show that the expected score of truth-telling is strictly higher than that of any no-effort strategy.

A no-effort strategy is essentially a mixture of two types of strategy. First, $\mu_i(X_i, Z_i)$ is independent of both $X_i$ and $Z_i$. In this case, we have shown in the proof of Theorem 4.3 that the expected score is 0, strictly less than the expected score of truth-telling, which is $I_{\text{TVD}}(X_i; X_j \mid Z)$.

Second, $\mu_i(X_i, Z_i)$ depends only on $Z_i$; let $\mu_i(X_i, Z_i) = \hat{Z}_i$. Conditioned on $Z$, the Markov chain $Z_i \to \hat{Z}_i$ holds. Now, fix agent $j$'s lazy-reporting strategy and let her report be $\hat{X}_j$. By the data processing inequality Shannon [1948], we have:

$$I_{\text{TVD}}(\hat{Z}_i; \hat{X}_j \mid Z) \le I_{\text{TVD}}(Z_i; \hat{X}_j \mid Z).$$

Next, suppose agent $i$ reporting $Z_i$, which is the score-maximizing lazy-reporting strategy regardless of agent $j$'s strategy. For any lazy-reporting strategy such that $\hat{X}_j = \nu_j(X_j, Z_j)$ that agent $j$ plays, applying the data processing inequality to agent $j$'s report yields:

$$I_{\text{TVD}}(Z_i; \hat{X}_j \mid Z) \le \max\left( I_{\text{TVD}}(Z_i; Z_j \mid Z), I_{\text{TVD}}(Z_i; X_j \mid Z) \right).$$

Therefore, Assumption 4.5 is sufficient for information monotonicity under lazy-reporting strategies.
$\square$

## E   Learning the Scoring Function

In Section 4.1, we assumed that the underlying scoring function $T_\Delta$ is known. For complex signal structures, especially when the signal space is large, we have to learn the scoring function from the data. This introduces a concern: if a significant fraction of agents rely on LLMs, the learned scoring function $T_{\tilde{\Delta}}$ might be manipulated in a way that benefits the LLM-generated reports. We mitigate this concern using the following proposition. Let $S_i^T(\theta_i, \theta_j)$ be the expected score computed under the scoring function $T$ when agents' strategies are $\theta_i$ and $\theta_j$ respectively.

**Proposition E.1.** *Suppose Assumption 4.1 and Assumption 4.2 hold with no error such that $\epsilon = \epsilon_i = 0$ for any $i \in [n]$. Then, for any scoring function $T \in \{0,1\}^{|\Sigma| \times |\Sigma|}$ and any strategy profile $(\theta_i, \theta_j)$, $S_i^T(\theta_i, \theta_j) \le S_i^{T_\Delta}(\tau, \tau)$, where $\tau$ denote the truth-telling strategy.*

Intuitively, the proposition holds because when $T_\Delta$ is known and agents are truth-telling, the expected score is $I_{\text{TVD}}(X_i; X_j \mid Z)$. However, if $T_\Delta$ is mislearned, the expected score is upper bounded by $I_{\text{TVD}}(\hat{X}_i; \hat{X}_j \mid Z)$. Since our assumptions ensure that $Z_i$ and $Z_j$ are independent given $Z$, the data processing inequality guarantees that $I_{\text{TVD}}(\hat{X}_i; \hat{X}_j \mid Z) < I_{\text{TVD}}(X_i; X_j \mid Z)$.

Proposition E.1 implies that the expected score of the conditioned CA mechanism is maximized if every agent exerts full effort and reports $X_i$, while any strategy that manipulates the true scoring function will only (weakly) decrease the expected score.[3]

*Proof of Proposition E.1.* Note that when agent $i$ and $j$ play strategy $\theta_i$ and $\theta_j$ respectively and the scoring function is $T$,

$$S_i^T(\theta_i, \theta_j)$$
$$= \sum_{z \in \Sigma} \Pr(Z = z) \sum_{x_i, x_j, z_i, z_j \in \Sigma} (\Pr(x_i, x_j, z_i, z_j \mid z) - \Pr(x_i, z_i \mid z) \Pr(x_j, z_j \mid z)) \, T(\theta_i(x_i, z_i), \theta_j(x_j, z_j), z)$$

Because Assumption 4.2 and Assumption 4.1 hold with no error, by Lemma D.2 and Lemma D.1, it is easy to show that $\Pr(x_i, x_j, z_i, z_j \mid z) = \Pr(x_i, x_j \mid z) \Pr(z_i \mid z) \Pr(z_j \mid z)$ and $\Pr(x_i, z_i \mid z) = \Pr(x_i \mid z) \Pr(z_i \mid z)$.

$$= \sum_{z \in \Sigma} \Pr(Z = z) \sum_{z_i, z_j \in \Sigma} \Pr(z_i \mid z) \Pr(z_j \mid z)$$
$$\sum_{x_i, x_j \in \Sigma} (\Pr(x_i, x_j \mid z) - \Pr(x_i \mid z) \Pr(x_j \mid z)) \, T(\theta_i(x_i, z_i), \theta_j(x_j, z_j), z)$$

$$\leq \sum_{z \in \Sigma} \Pr(Z = z) \sum_{z_i, z_j \in \Sigma} \Pr(z_i \mid z) \Pr(z_j \mid z) \sum_{x_i, x_j \in \Sigma} |\Pr(x_i, x_j \mid z) - \Pr(x_i \mid z) \Pr(x_j \mid z)|$$
$$\text{(Because } T(k, h, l) \text{ is either 0 or 1.)}$$

$$= \sum_{z \in \Sigma} \Pr(Z = z) \sum_{x_i, x_j \in \Sigma} |\Pr(x_i, x_j \mid z) - \Pr(x_i \mid z) \Pr(x_j \mid z)| \qquad \text{(Marginalize } z_i, z_j)$$

$$= S_i^{T_\Delta}(\tau, \tau)$$

$\square$

# F  A Heuristic Generalization to Complex Signals

We have focused on scenarios where the signal space is small, such as answering multi-choice questions like sentiment labeling, preference elicitation, or grading. However, many real-world crowdsourcing tasks involve open-ended questions that require workers to provide more complex responses, such as image or video descriptions, idea generation, or feedback and review collection. These tasks introduce unique challenges for applying our method, including 1) defining meaningful representation (textual) responses, and 2) computing the information between two responses conditioned on the principal's sample $Z$. In this section, we outline how to address these challenges and extend our method to accommodate tasks with complex, open-ended responses. We will illustrate our idea using peer review as an example, while we note that our method can be generalized to other similar settings.

**Representing responses.**    Suppose two agents review the same paper and each has a review after reading the paper, which is denoted as $X_i$ and $X_j$ respectively. Instead of working hard to obtain a high-quality review, each agent can use an LLM to generate a fake review, denoted as $Z_i$ and $Z_j$. The first challenge is how to measure the level of agreement between two reviews. The idea is to represent each review with an embedding vector and compute the similarity between a pair of embedding vectors. Depending on the information of interest, different embedding methods can be applied. For example, classic NLP embedding methods such as Word2Vec Mikolov [2013], TF-IDF Manning [2009], BERT and its variants Devlin [2018], Sanh [2019], and GPT-3 Brown et al. [2020] can be used to represent the language-level information within the responses.

Existing embedding methods may struggle when the focus is on content-level information, such as the judgments in peer reviews. For example, at the language level, an LLM-generated review may be similar to a human-written review that is rewritten for polishing by an LLM, while the information within their judgments can be very different. To address this, we consider an alternative, interpretable

---

[3]Here we assume that the principal can learn the correct $T_\Delta$ if everyone reports their full-effort signal.

approach for representing complex responses. The key idea is to prompt an LLM to score a response based on a list of pre-determined characteristics.

For instance, we can prompt an LLM to classify whether a review adopts a compromising, critical, or neutral stance on specific aspects of a paper, such as writing quality, completeness of related work, novelty of the research idea, or the quality of datasets. Using this method, each textual review $X$ can be represented as a vector $\phi(X)$ of length $K$, where each entry $\phi(X)_k$ encodes the attitude of the review toward the $K$-th aspect. The similarity between two reviews $X_i$ and $X_j$ can then be quantified using the cosine similarity between their representations $\phi(X_i)$ and $\phi(X_j)$.

The same idea can be generalized beyond peer review. Suppose the principal has a pre-defined set of questions about the responses, denoted as $\mathcal{Q}$ where $|\mathcal{Q}| = K$. Each question requires a (normalized) score between $-1$ and $1$, ensuring that the inner product between vectors can serve as a meaningful similarity measure. Given a response $X_i$, the principal can prompt an LLM to iteratively get an answer to each question $q \in \mathcal{Q}$. This process converts each response $X_i$ into a vector $\phi(X_i)$ of length $K$, and each entry is a score between $-1$ and $1$. Similar ideas have been applied to identify similar data points Zeng et al. [2024], analyze peer review Liang et al. [2024], and select features Jeong et al. [2024].

**Computing the conditioned similarity.**  The key idea of the conditioned CA mechanism is to give an agent $i$ a higher score for having responses that correlate with another agent $j$'s responses beyond what both share with $Z$. To measure how $\phi(X_i)$ and $\phi(X_j)$ correlate conditioned on $\phi(Z)$, a direct application—treating each vector component as a random variable—quickly becomes infeasible due to the exponential growth of the signal space in the vector dimension $K$. Therefore, we adopt a heuristic: we compute the similarity between $\phi(X_i)$ and $\phi(X_j)$ after removing their shared direction with $\phi(Z)$. Specifically:

1. **Project each vector onto the hyperplane orthogonal to $\phi(Z)$.**
   Let $u_z = \frac{\phi(Z)}{\|\phi(Z)\|}$ be the unit vector in the direction of $\phi(Z)$. Then,
   $$\phi(X_i)' = \phi(X_i) - (\phi(X_i) \cdot u_z)u_z, \qquad \phi(X_j)' = \phi(X_j) - (\phi(X_j) \cdot u_z)u_z.$$

2. **Compute the cosine similarity** between $\phi(X_i)'$ and $\phi(X_j)'$ to score agent $i$.

By projecting the embedding vectors onto the orthogonal complement of $\phi(Z)$, we remove the information that already exists in $Z$. This projection approach is also used in NLP for "debiasing" word embeddings by eliminating similarity along a "biased direction" Bolukbasi et al. [2016].

Finally, we note that we do not assume the principal's questions in $\mathcal{Q}$ to be i.i.d. Thus, the theoretical guarantees in Section 4.1 do not directly apply to this heuristic generalization; we instead rely on empirical evaluation to assess performance.

## G  Experiments (Continue)

### G.1  Large Language Models

In this work, we select five well-known LLMs to generate labels or annotations for all used datasets, including GPT-3.5-turbo, GPT-4, Gemma-2-2b-it[4], Phi-3.5-mini-Instruct [5], and Mistral-7B-Instruct-v0.3[6]. The corresponding prompt templates are provided in Appendix H. Note that we choose only five models because other commonly used LLMs such as LLaMA and Gemini refuse to answer toxicity-related questions.

### G.2  Details of the Baseline Methods

**Output Agreement (OA)**  Perhaps the most straightforward idea to score an agent is to count the frequency of agreements between her responses and another agents' responses. Specifically, while scoring agent $i$, we iteratively pair her with a different agent $j$ and let $M_{i,j}$ denote the set of questions

---

[4]https://huggingface.co/google/gemma-2-2b-it
[5]https://huggingface.co/microsoft/Phi-3.5-mini-instruct
[6]https://huggingface.co/mistralai/Mistral-7B-Instruct-v0.3

answered by both agents. Then, the score of agent $i$ is the average of the empirical probability of agreement, i.e. $\frac{1}{n}\sum_{j\in[n]\setminus\{i\}}\sum_{q\in M_{i,j}}\frac{\mathbb{1}[\hat{X}_{i,q}=\hat{X}_{j,q}]}{|M_{i,j}|}$ where $\hat{X}$ is the crowdsourced dataset with rows representing agents and columns representing questions.[7] A score of 1 thus indicates perfect agreement while a score of 0 indicates no agreement at all.

**Correlated Agreement (CA)** We use the CA mechanism Shnayder et al. [2016] as a baseline which does not incorporate the principal's information $Z$. Details of the mechanism have been presented in Algorithm 1.

**Expectation-Maximization (EM)** The EM algorithm is a well-known approach to estimating the reliabilities of noisy agents and better recovering the ground truth considering the reliabilities of agents [Dawid and Skene, 1979]. Each agent $i$ is assumed to make mistakes according to a confusion matrix $\Gamma^i$ where $\Gamma^i_{h,l}$ denotes the probability that agent $i$ reports $h$ when the ground truth answer is $l$. The EM algorithm first initializes a prior of the ground truth for each question and a confusion matrix for each agent. Then, it iteratively goes through an E-step and an M-step. In the E-step, the algorithm computes the posterior of the ground truth of each question given the prior and the confusion matrix. In the M-step, the confusion matrix of each agent is updated using the posterior computed in the E-step. The above two steps are iterated until the likelihood of $D$ converges. Finally, we score the reliability of agent $i$ as $\sum_{h\in\Sigma}\Gamma^i_{h,h}\tilde{P}(h)$ where $\tilde{P}(h)$ is the empirical marginal distribution of observing $h$ in the dataset.

**Conditioned Agreement ($\mathrm{OA}_Z$)** The first three measures do not utilize the principal's information about the cheap signal, denoted as $Z$. Note that in our experiments, $Z$ is the vector of labels generated by a particular LLM. The conditioned agreement metric scores a agent if she can provide agreements to another agent in addition to $Z$. Similar to the agreement metric, the score of agent $i$ is the average of the empirical probability of agreement conditioned on $Z$, i.e. $\frac{1}{n}\sum_{j\in[n]\setminus\{i\}}\sum_{q\in M_{i,j}}\frac{\mathbb{1}[\hat{X}_{i,q}=\hat{X}_{j,q}\,\&\,\hat{X}_{i,q}\neq Z_q]}{|M_{i,j}|}$. Intuitively, if agent $i$ uses an LLM whose generated labels are similar to $Z$, then $\hat{X}_{i,q}\neq Z_q$ will happen with a small probability, meaning that the conditioned agreement score will be small.

## G.3 Assumption Verifications (Continue)

Table 2, Table 3, Table 4 and Table 5 present the mutual information $I_{\mathrm{TVD}}(Z_i; Z_j \mid Z)$ on four datasets. $Z$ and $Z_i$ are sampled by the same LLM indicated on the rows and $Z_j$ is sampled by the LLM indicated on the columns. The first observation is that all values displayed in black exceed 0.05, which represents the mutual information of the random reporting baseline. This implies that $I_{\mathrm{TVD}}(Z_i; Z_j \mid Z) > \epsilon$ holds consistently, rejecting hypothesis 2.

Regarding hypothesis 4, we observe that the average correlations between two LLMs are generally weaker than those between two human agents, provided $Z_i$ and $Z$ are independently generated by the same model. This trend is evident in the tables, where almost all bracketed values are positive, with one exception.[8]

## G.4 Detecting Low-effort Agents (Continue)

### G.4.1 Additional Figures and Tables

In Figure 4, we present additional figures on the toxicity dataset, analogous to Figure 3b, for other LLMs. Analogous results to Table 1 for other datasets are summarized in Table 6, Table 7, Table 8, and Table 9. These results support our main claim that, although our method is occasionally slightly outperformed by certain baselines, it consistently demonstrates the most robust performance across all tested settings.

---

[7]Let $\frac{0}{0} = 0$.

[8]The exception occurs in the preference alignment dataset, where samples of $Z_i$, $Z_j$, and $Z$ all generated by GPT-4.

Table 2: Conditional TVD Mutual Information Between Two LLMs on the Preference Alignment Data

|  | GPT-3.5 | GPT-4 | Gemma | Mistral | Phi |
|---|---|---|---|---|---|
| GPT-3.5 | 0.2255 (0.0412) | 0.1973 (0.0694) | 0.0865 (0.1802) | 0.1224 (0.1443) | 0.1311 (0.1355) |
| GPT-4 | 0.0715 (0.1537) | 0.2559 (-0.0307) | 0.0571 (0.1682) | 0.0552 (0.1701) | 0.0723 (0.1530) |
| Gemma | 0.1013 (0.1819) | 0.1767 (0.1066) | 0.2453 (0.0379) | 0.0655 (0.2177) | 0.1174 (0.1659) |
| Mistral | 0.0920 (0.1938) | 0.1132 (0.1726) | 0.0620 (0.2238) | 0.2350 (0.0508) | 0.0964 (0.1894) |
| Phi | 0.0874 (0.1969) | 0.1300 (0.1542) | 0.0855 (0.1988) | 0.0929 (0.1914) | 0.2479 (0.0364) |

Each entry represents $I_{\text{TVD}}(Z_i; Z_j \mid Z)$ (Difference: $I_{\text{TVD}}(X_i; X_j \mid Z) - I_{\text{TVD}}(Z_i; Z_j \mid Z)$), where $Z_i$ and $Z_j$ are generated by the models in the row and column respectively, and $Z$ uses the same model as $Z_i$.

Table 3: Conditional TVD Mutual Information Between Two LLMs on the Hatefulness Labeling Data

|  | GPT-3.5 | GPT-4 | Gemma | Mistral | Phi |
|---|---|---|---|---|---|
| GPT-3.5 | 0.0317 (0.1605) | 0.0362 (0.1560) | 0.0385 (0.1537) | 0.0426 (0.1496) | 0.0353 (0.1570) |
| GPT-4 | 0.0077 (0.1575) | 0.0286 (0.1366) | 0.0229 (0.1424) | 0.0216 (0.1437) | 0.0253 (0.1400) |
| Gemma | 0.0075 (0.1588) | 0.0183 (0.1481) | 0.0758 (0.0905) | 0.0464 (0.1199) | 0.0543 (0.1121) |
| Mistral | 0.0040 (0.1600) | 0.0097 (0.1542) | 0.0341 (0.1299) | 0.0408 (0.1231) | 0.0350 (0.1290) |
| Phi | 0.0088 (0.1658) | 0.0115 (0.1631) | 0.0618 (0.1128) | 0.0526 (0.1220) | 0.0803 (0.0943) |

### G.4.2 Detection Performance vs. Number of Samples

We further examine how many questions the principal needs to sample to effectively detect low-effort agents. In Figure 5, we report the AUC of the conditioned CA mechanism with the fraction of LLM-reliant agents fixed at 0.15, while the fractions of random and biased agents are independently sampled from [0, 0.2]. We vary the sampling rate from 20% to 100%. The results vary across the datasets and the LLMs that agents use. Except for the hatefulness labeling dataset with GPT-3.5-reliant agents, sampling 50% of the questions is sufficient to outperform the baseline CA mechanism.

### G.4.3 Conditioning on samples of multiple LLMs

In practice, agents may employ a variety of LLMs for low-effort response generation, and the principal lacks prior knowledge of which models they are using. To address this challenge, we propose an iterative conditioning approach, systematically removing the influence of responses generated by each LLM available to the principal. The idea is to iteratively compute the CA score conditioned on each of the LLM samples the principal has, and then take the minimum as the final score. Intuitively, human agents will perform consistently well when conditioning out any of the LLMs' information. However, LLM-reliant agents will perform poorly in at least one of the iterations and thus leading to a lower final score.

In Figure 6, we replicate Figure 3b for the setting where the LLM-reliant agents randomly pick two LLMs with half-half probability while the principal iteratively conditions out his samples for these two LLMs' answers. Our method exhibits the most robust performance across all datasets, echoing the previous conclusion.

### G.5 Complex Signals

We further test the idea illustrated in Appendix F that generalizes our method to high-dimensional settings.

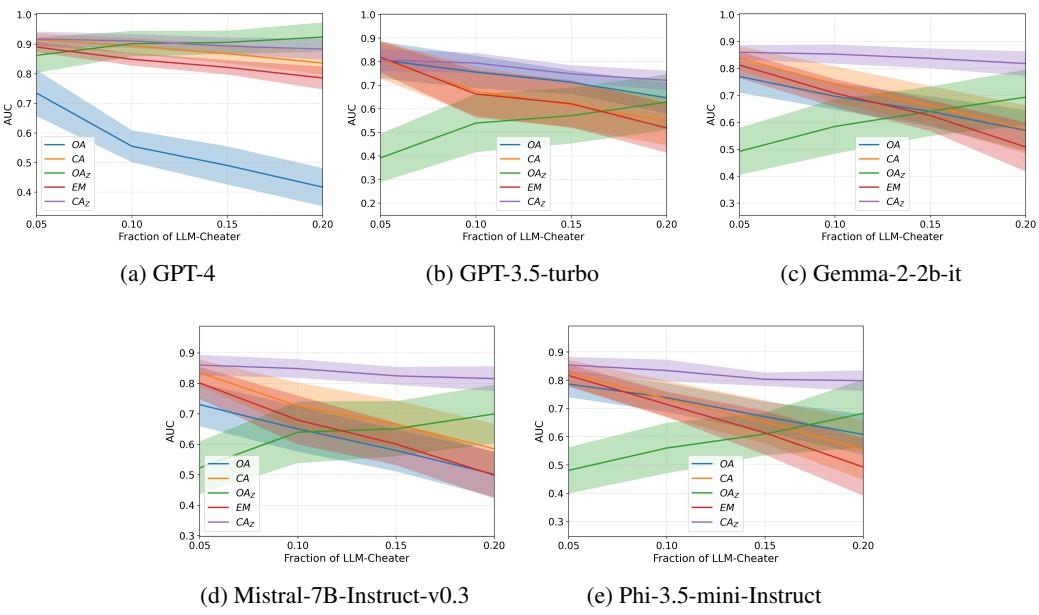

(a) GPT-4  (b) GPT-3.5-turbo  (c) Gemma-2-2b-it

(d) Mistral-7B-Instruct-v0.3  (e) Phi-3.5-mini-Instruct

Figure 4: The area under the ROC curve (AUC) for various methods while varying the fraction of LLM-reliant agents. These are the analogous figures for Figure 3b for the remaining LLMs on the toxicity labeling dataset.

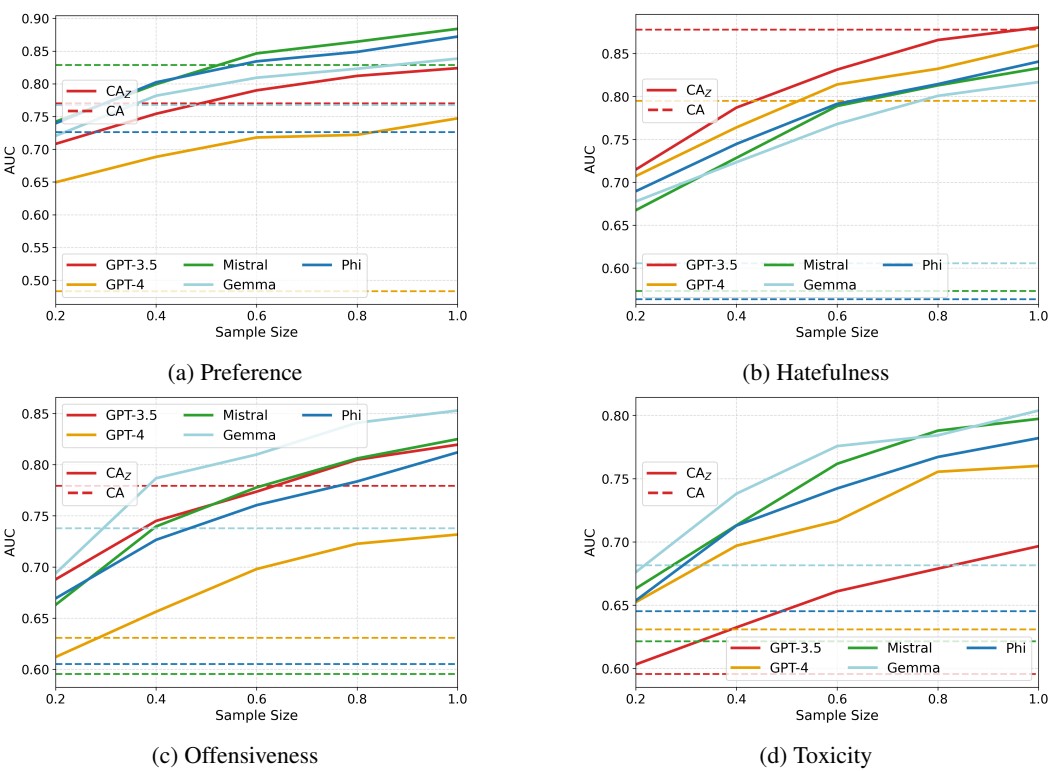

(a) Preference  (b) Hatefulness

(c) Offensiveness  (d) Toxicity

Figure 5: The area under the ROC curve (AUC) for various methods while varying the fraction of questions the principal can sample.

Table 4: Conditional TVD Mutual Information Between Two LLMs on the Offensiveness Labeling Data

|  | GPT-3.5 | GPT-4 | Gemma | Mistral | Phi |
|---|---|---|---|---|---|
| GPT-3.5 | 0.1267 (0.0809) | 0.0926 (0.1149) | 0.0450 (0.1625) | 0.0785 (0.1290) | 0.0802 (0.1273) |
| GPT-4 | 0.0419 (0.1092) | 0.0920 (0.0590) | 0.0520 (0.0990) | 0.0867 (0.0644) | 0.0794 (0.0716) |
| Gemma | 0.0302 (0.2336) | 0.0602 (0.2036) | 0.1152 (0.1486) | 0.0597 (0.2040) | 0.0676 (0.1962) |
| Mistral | 0.0390 (0.1407) | 0.0573 (0.1225) | 0.0295 (0.1502) | 0.0836 (0.0961) | 0.0516 (0.1281) |
| Phi | 0.0320 (0.1604) | 0.0597 (0.1327) | 0.0334 (0.1591) | 0.0604 (0.1321) | 0.0651 (0.1273) |

Table 5: Conditional TVD Mutual Information Between Two LLMs on the Toxicity Labeling Data

|  | GPT-3.5 | GPT-4 | Gemma | Mistral | Phi |
|---|---|---|---|---|---|
| GPT-3.5 | 0.1611 (0.0063) | 0.1328 (0.0347) | 0.1199 (0.0476) | 0.1419 (0.0255) | 0.1340 (0.0334) |
| GPT-4 | 0.0587 (0.1399) | 0.1035 (0.0951) | 0.0755 (0.1231) | 0.0673 (0.1312) | 0.0652 (0.1334) |
| Gemma | 0.0721 (0.1650) | 0.0869 (0.1502) | 0.1329 (0.1042) | 0.0880 (0.1491) | 0.1043 (0.1328) |
| Mistral | 0.0602 (0.1413) | 0.0622 (0.1394) | 0.0454 (0.1561) | 0.0934 (0.1081) | 0.0642 (0.1373) |
| Phi | 0.0416 (0.1756) | 0.0517 (0.1655) | 0.0382 (0.1790) | 0.0472 (0.1701) | 0.0666 (0.1507) |

In this experiment, we randomly sample 500 papers from the ICLR 2024 OpenReview data and select three human reviews per paper as human responses $X$. For each of the same batch of papers, we further use each of the five LLMs to generate three AI reviews using three similar-meaning yet different prompts (Appendix H). We randomly pick one of the generated reviews as the principal's sample $Z$, and the other two generated reviews will be used to simulate the responses of LLM-reliant agents.

We prompt GPT-4 to process each human review and LLM-generated review into a 19-dimension vectors with the prompt shown in Appendix H. We further simulate a review embedding tensor with a size of $n \times m \times 19$, where there are $n = 188$ reviewers and $m = 500$ papers in total. Each reviewer randomly reviews $k \in \{5, 8\}$ papers. An $\alpha_{llm}$ fraction of reviewers are LLM-reliant agents whose review vectors are replaced with a particular type of LLM-generated reviews. An $\alpha_p$ fraction of reviewers are prior-reporting agents, where the score in each entry of the review vector is randomly sampled according to the marginal distribution in the dataset. An $\alpha_r$ fraction of reviewers are random agents, where the score in each entry of the review vector is sampled uniformly at random. In our experiments, we independently sample each of $\alpha_{llm}$, $\alpha_p$, and $\alpha_r$ uniformly from the interval $[0, 0.1]$ and take average over 100 trials. While constructing the review embedding tensor, we guarantee

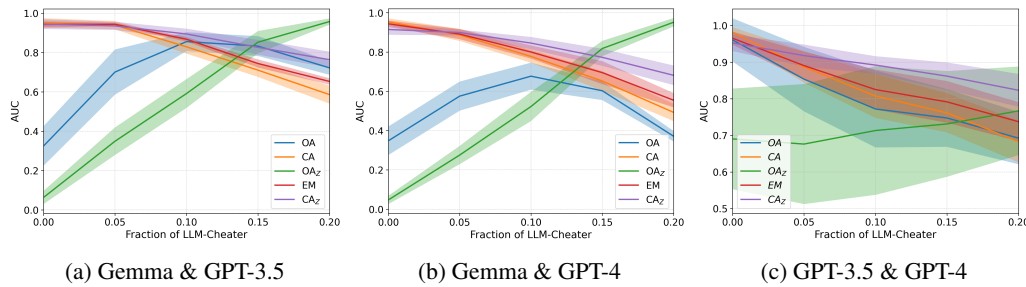

(a) Gemma & GPT-3.5  (b) Gemma & GPT-4  (c) GPT-3.5 & GPT-4

Figure 6: Area under the ROC curve (AUC) for various methods as the fraction of LLM-reliant agents increases. In each case, LLM-reliant agents randomly choose between two LLMs to generate responses, and the score is defined as the minimum of the two conditioned CA scores, each conditioned on samples from one LLM.

Table 6: Average AUC (and bottom 10% quantiles) across datasets and methods with GPT-3.5-reliant agents.

| Dataset | OA | CA | $\text{OA}_Z$ | EM | $\text{CA}_Z$ (ours) |
|---|---|---|---|---|---|
| Hatefulness | 0.61 (0.40) | **0.90** (0.84) | **0.90** (0.85) | 0.86 (0.77) | 0.89 (**0.86**) |
| Offensiveness | 0.80 (0.69) | 0.76 (0.58) | 0.48 (0.27) | 0.76 (0.57) | **0.83** (**0.77**) |
| Toxicity | 0.75 (0.62) | 0.72 (0.51) | 0.47 (0.24) | 0.71 (0.50) | **0.78** (**0.71**) |
| Preference Alignment | 0.88 (0.67) | 0.86 (0.69) | 0.88 (0.76) | **0.96** (**0.91**) | 0.91 (0.83) |

Table 7: Average AUC (and bottom 10% quantiles) across datasets and methods with Mistral-reliant agents.

| Dataset | OA | CA | $\text{OA}_Z$ | EM | $\text{CA}_Z$ (ours) |
|---|---|---|---|---|---|
| Hatefulness | 0.82 (0.75) | 0.72 (0.51) | 0.62 (0.48) | 0.76 (0.54) | **0.84** (**0.78**) |
| Offensiveness | 0.71 (0.54) | 0.79 (0.62) | 0.63 (0.47) | 0.75 (0.57) | **0.89** (**0.84**) |
| Toxicity | 0.65 (0.50) | 0.75 (0.56) | 0.58 (0.39) | 0.70 (0.49) | **0.84** (**0.79**) |
| Preference Alignment | 0.86 (0.66) | 0.89 (0.75) | **0.94** (0.87) | **0.94** (0.87) | 0.93 (**0.88**) |

that each paper has approximate 3 reviews so that the variance of the number of reviewers per paper is controlled. Again, we consider five types of LLMs and for each type of LLM-reliant agent, the principal's sample $Z$ is an independent generation of the same LLM under a different prompt.

We consider three methods to score the reviewers.

- **Cosine Similarity**: Scores each reviewer based on the average cosine similarity between their review vector and the review vectors of other reviewers for the same paper.

- **Projected Cosine Similarity**: Applies the method described in Appendix F, where each review vector is projected onto the subspace orthogonal to the principal reviewer's sampled embedding before computing similarity.

- **Negative Similarity to $Z$**: Scores each reviewer by taking the negative average cosine similarity between their review vector and the principal reviewer's sampled vector $Z$.

We present the results in Figure 7. It is clear that conditioning out the principal's sampled LLM reviews is more effective than computing the cosine similarity directly. Furthermore, naively removing reviews that are similar to $Z$ has a large variance, whose performance greatly depends on the existence of random agents and prior-reporting agents.

We further note that there is a big room for improvement. First, from a methodology perspective, the 19-dimensional embedding is heuristically chosen and is not optimized for distinguishing human-AI reviews. Second, from the data perspective, human reviews are naturally very noisy. Our method can be potentially applied to other crowdsourcing tasks where human responses are more consistent. We view these as future directions.

# H   Prompts Templates

We adopt our prompt templates based on recent studies [Pang et al., 2024, Liu et al., 2024, Pang et al., 2025]. The specific templates used for the two dataset settings are presented in Table H and Table H, respectively.

Table 8: Average AUC (and bottom 10% quantiles) across datasets and methods with Gemma-reliant agents.

| Dataset | OA | CA | $OA_Z$ | EM | $CA_Z$ (ours) |
|---|---|---|---|---|---|
| Hatefulness | **0.85** (0.79) | 0.71 (0.51) | 0.60 (0.42) | 0.77 (0.56) | **0.85** (**0.80**) |
| Offensiveness | 0.75 (0.58) | 0.78 (0.59) | 0.61 (0.43) | 0.76 (0.53) | **0.90** (**0.84**) |
| Toxicity | 0.70 (0.58) | 0.75 (0.59) | 0.56 (0.36) | 0.72 (0.52) | **0.85** (**0.79**) |
| Preference Alignment | 0.80 (0.58) | 0.87 (0.73) | 0.76 (0.62) | **0.92** (0.85) | **0.92** (**0.86**) |

Table 9: Average AUC (and bottom 10% quantiles) across datasets and methods with Phi-reliant agents.

| Dataset | OA | CA | $OA_Z$ | EM | $CA_Z$ (ours) |
|---|---|---|---|---|---|
| Hatefulness | **0.87** (**0.81**) | 0.67 (0.44) | 0.66 (0.47) | 0.77 (0.56) | 0.85 (0.79) |
| Offensiveness | 0.76 (0.64) | 0.76 (0.59) | 0.58 (0.35) | 0.75 (0.53) | **0.88** (**0.83**) |
| Toxicity | 0.73 (0.62) | 0.74 (0.54) | 0.54 (0.35) | 0.71 (0.48) | **0.83** (**0.77**) |
| Preference Alignment | 0.85 (0.68) | 0.82 (0.63) | 0.82 (0.63) | **0.97** (**0.94**) | 0.93 (0.88) |

---

**Prompt Template for Toxicity Labeling**

**System Prompt**: As a helpful system judgment assistant, your task is to answer the following questions for the given comment. You should label a comment as **hateful** if it expresses hatred towards a targeted group or is intended to be derogatory, to humiliate, or to insult the members of the group. You should label a comment as **offensive** if it contains hurtful, derogatory, or obscene comments. You should label a comment as **toxic** if it is a rude, disrespectful, or unreasonable comment that is likely to make readers want to leave a discussion.

**Question 1**: Do you think this comment is hateful? —Yes or No.

**Question 2**: Do you think this comment is offensive? —Yes or No.

**Question 3**: Do you think this comment is toxic? —Yes or No.

**Comment**: `[Comment]`

**User Prompt**: Now, please evaluate the following comment and return the final answers in JSON format:

```
{
"question 1": {"hateful": <yes/no>},
"question 2": {"offensive": <yes/no>},
"question 3": {"toxic": <yes/no>}
}
```

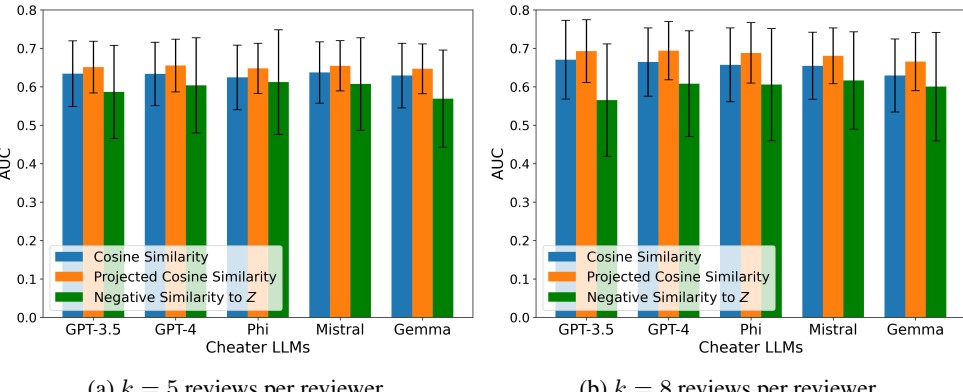

(a) $k = 5$ reviews per reviewer  (b) $k = 8$ reviews per reviewer

Figure 7: A comparison of the area under the ROC curve (AUC) of various methods in the peer review dataset.

## Prompt Template for Preference Alignment

**System Prompt**:
Please act as an impartial judge and evaluate the quality of the responses provided by two AI assistants to the user question displayed below. You should choose the assistant that follows the user's instructions and answers the user's question better. Your evaluation should consider factors such as the **helpfulness, harmlessness, truthfulness**, and **level of detail** of their responses. Begin your evaluation by comparing the two responses and provide a short explanation. Avoid any position biases and ensure that the order in which the responses were presented does not influence your decision. Do not allow the length of the responses to influence your evaluation. Do not favor certain names of the assistants. Be as objective as possible.

After providing your explanation, output your final verdict by strictly following a **5-point Likert scale**:
- **A is clearly better**
- **A is slightly better**
- **Tie**
- **B is slightly better**
- **B is clearly better**

**Question**: [Question]
**Response A**: [Response A]
**Response B**: [Response B]

**User Prompt**:
Now, please evaluate the responses and return the final verdict in JSON format:

```
{
    "final verdict": <A-is-clearly-better / A-is-slightly-better, Tie,
    B-is-slightly-better / B-is-clearly-better>,
    "short explanation": <your short explanation within one sentence>
}
```

## Prompt Template for Paper Review 19-Dimensional Judgment Embedding Generation

**System Prompt**:
You will be given a review of a computer science paper. Such a review is typically composed of a summary of the corresponding paper, reasons to accept, reasons to reject, and questions for the authors. Upon receiving the review, your task is to generate an embedding of the review in a 19-dimensional space according to the following instructions:

1. Identify and ignore the summary and the questions section of the review (if any).
2. Separate the review into advantages and disadvantages.
3. Further decompose the advantages and disadvantages into independent judgments.
4. You will be given **19 judgment categories**. For each category:
     - If a positive-toned judgment falls into the category, assign a score of 1.
     - If a negative-toned judgment falls into the category, assign a score of -1.
     - If no judgment is related to the category, assign a score of 0.
5. Return the scores for all 19 categories as a dictionary. Do not include any additional explanation.

The judgment categories are:

| | | |
|---|---|---|
| Clarity and Presentation | Related Work | Theory Soundness |
| Experiment Design | Significance of the Problem | Novelty of Method |
| Practicality of Method | Comparison to Previous Studies | Implications of the Research |
| Reproducibility | Ethical Aspects | Relevance to Conference |
| Experiment Execution | Scope and Generalizability | Utility and Applicability |
| Defense Effectiveness | Real-world Applicability | Algorithm Efficiency |
| Data Generation and Augmentation | | |

**User Prompt**:
Now, please evaluate the paper review and return the scores for each category in the following JSON format:

```
{
    "Clarity and Presentation": <-1/0/1>,
    "Related Work": <-1/0/1>,
    "Theory Soundness": <-1/0/1>,
    "Experiment Design": <-1/0/1>,
    "Significance of the Problem": <-1/0/1>,
    "Novelty of Method": <-1/0/1>,
    "Practicality of Method": <-1/0/1>,
    "Comparison to Previous Studies": <-1/0/1>,
    "Implications of the Research": <-1/0/1>,
    "Reproducibility": <-1/0/1>,
    "Ethical Aspects": <-1/0/1>,
    "Relevance to Conference": <-1/0/1>,
    "Experiment Execution": <-1/0/1>,
    "Scope and Generalizability": <-1/0/1>,
    "Utility and Applicability": <-1/0/1>,
    "Defense Effectiveness": <-1/0/1>,
    "Real-world Applicability": <-1/0/1>,
    "Data Generation and Augmentation": <-1/0/1>,
    "Algorithm Efficiency": <-1/0/1>
}
```

