# OpenReview forum: "Evaluating LLM-contaminated Crowdsourcing Data Without Ground Truth"
_NeurIPS.cc/2025/Conference — NeurIPS 2025 poster_

### Official Review · Reviewer_DdeM · 2025-06-09

**Clarity:** 4
**Significance:** 3
**Originality:** 3
**Rating:** 5
**Confidence:** 3

**Summary:**

This research aims to address the impact of Large Language Model (LLM)-assisted cheating on the data quality of crowdsourcing. Traditional methods for dealing with low-quality workers in crowdsourcing often rely on correlations among workers' responses. However, such methods are ill-suited for situations where correlations arise from the use of LLM-generated outputs.

To overcome this, this study adopts an approach that utilizes correlations between workers' responses, conditioned on LLM-generated outputs. By doing so, the paper proposes a theoretically grounded quality measure for crowdsourced annotation tasks. The effectiveness of this approach is further evaluated through empirical experiments.

**Questions:**

N/A

**Ethical Concerns:**

["NO or VERY MINOR ethics concerns only"]

**Final Justification:**

My evaluation of this paper has not changed.

**Limitations:**

yes

**Quality:**

4

**Strengths And Weaknesses:**

This paper appears to be very well-written in my assessment. The motivation is clear, and I believe the content is a good fit for this conference.
A weakness of this paper is that its primary focus is on crowdsourcing datasets involving subjective questions or tasks where human experts are considered the gold standard. With the recent advancements in LLMs, the scope of applicable tasks might decrease over time.

Nevertheless, overall, I consider this to be a good paper.

---

> ### Author Rebuttal · Authors · 2025-07-30
>
> Thank you for your appreciation! We agree that human-based crowdsourcing is gradually being replaced (at least reshaped) by AI, and may eventually be fully replaced. This means the value of the current paper may be timely discounted (and we hope so). However, at least for the next couple of years, we envision that protecting genuine human feedback will still be an essential and important issue, such as supervising trustworthy AI via RLHF. Now is perhaps the right time to pay more attention to this pressing problem.

---

> > ### Comment · Reviewer_DdeM · 2025-08-06
> >
> > Thank you for your rebuttal comment. I look forward to further discussion on this important issue.

---

> > > ### Author Response · Authors · 2025-08-06
> > >
> > > Thank you for the follow-up. We envision that human effort will continue to play a central role in the development of trustworthy, safe, and aligned AI. While recent advances in LLMs and techniques such as RLAIF or human-AI delegation can help reduce reliance on human effort for routine tasks, these approaches still fundamentally depend on high-quality human-generated data to anchor and guide the data collection or training process.
> > >
> > > It is true that for some objective tasks, LLMs may increasingly serve as strong initial annotators. However, in many settings — especially those involving ambiguous judgments or evolving societal norms — genuine human insight remains indispensable. Even as the boundary of "human-required" tasks shifts, the importance of incentivizing careful and honest human input does not diminish. In fact, as AI systems are deployed in more consequential settings, the need to ensure the integrity and thoughtfulness of human oversight becomes even more critical.
> > >
> > > Thus, we believe our focus on mechanisms that encourage genuine human effort is not only timely but foundational to future AI progress. We look forward to any further discussions.

---

### Official Review · Reviewer_huUj · 2025-06-26

**Clarity:** 2
**Significance:** 2
**Originality:** 2
**Rating:** 4
**Confidence:** 4

**Summary:**

This paper proposes theoretically grounded quality measure method for annotation tasks, called the conditioned correlated agreement (CA) mechanism. The method quantifies the correlations between worker answers while conditioning on (a subset of) LLM-generated labels available to the requester. Building on prior research, the proposed training-free scoring mechanism is theoretical guaranteed under a novel model that accounts for LLM collusion. Experimental results show that their method is effective in identifying LLM-assisted cheating when conditioning on the “right” model.

**Questions:**

Please refer to the weaknesses above.

**Ethical Concerns:**

["NO or VERY MINOR ethics concerns only"]

**Final Justification:**

The author has adopted my suggestions regarding Figures 2, 3, and 4, and provided additional explanations for the experimental metrics. Considering the feedback from other reviewers, I have decided to keep my score unchanged.

**Limitations:**

yes

**Quality:**

2

**Strengths And Weaknesses:**

Strengths：
1. Compared with existing LLM detection methods, the method proposed in this paper is based on peer prediction and is suitable for structured annotation tasks like multiple-choice labeling.

2. The proposed conditioned CA mechanism is information monotone under naive strategies, which motivates workers to truthfully report their high-effort signals.

3. Experiments demonstrates that the proposed conditioned CA mechanism is robust in detecting low-effort cheating on real-world crowdsourcing datasets.

Weaknesses:
1. Figure 2 only shows the results on (a) Hatefulness and (b) Preference Alignment. Why not directly show the results on (a) Hatefulness, (b) Offensiveness, (c) Toxicity and (d) Preference Alignment? As the extension of Figure 2, Figure 4 shows the results on Hatefulness again, but the result is different from Figure 2. Yet at the same time, the result on Toxicity is not shown. Please check if the figures are incorrectly labeled.

2. In Figure 2's legend, “Expert”, “Normal worker”, and “Human” refer to X_i according to the “Assumption Verification” subsection, which is inconsistent with the description of Figure 2’s caption.

3. In Figure 3b, it is not clear which methods the different colored lines correspond to respectively. Please provide a legend.

4. In subsection 5.3, the metric chosen to measure the effectiveness of proposed method to identify cheating agents with naive strategies is AUC. How does AUC reflect the effectiveness of detecting cheating agents?

---

> ### Author Rebuttal · Authors · 2025-07-30
>
> We appreciate your clarification questions about the figures. They all make great sense and will be adjusted in the future version.
> * Label mistake in Figure 4
>   * Thanks for pointing this out! Figure 4 (a) should be “toxicity” rather than “hatefulness”. We defer two figures to the appendix for space considerations.
> * Captain of Figure 2
>   * You are right. We will update the caption to “The TVD mutual information between $X_i$ and $X_j$ (for “Human”, “Normal worker”, and “expert”) or $Z_i$ and $X_j$ (for remaining bars) conditioned on $Z$.” We will also update the y-axis label.
> * Missing legend for Figure 3
>   * Thanks for pointing out the mistake. The legend is the same as Figure 5 (will update).
> * AUC curves
>   * In our context, let $I^+$ be the set of hard-working human labelers and $I^-$ be the set of simulated cheaters. Let $S_i$ be the score of labeler i. We are able to compute the AUC score as follows:
>  $$ \text{AUC} = \frac{1}{|I^+| \cdot |I^-|} \sum_{i \in I^+} \sum_{j \in I^-} (\mathbb{1}[ S_i > S_j ] + 0.5 \cdot\mathbb{1}[ S_i = S_j ])$$
>   * Intuitively, AUC quantifies the probability that a randomly chosen good labeler is ranked higher than a randomly chosen cheater under each scoring function. It measures how well each scoring function can be used to distinguish good workers from cheaters. We will further explain this concept in future versions of our paper.

---

> ### Author Response · Authors · 2025-08-05
> **Thank you for your feedback**
>
> Thank you for your review and appreciation of our work. We hope our responses have clarified all your questions. If you have anything further you'd like to discuss, we’d be happy to continue the conversation.

---

> > ### Comment · Reviewer_huUj · 2025-08-07
> >
> > Thank you for the clarifications. My concerns have been addressed. After considering the feedback from other reviewers, I will maintain my positive recommendation.

---

### Official Review · Reviewer_NyJA · 2025-07-01

**Clarity:** 3
**Significance:** 2
**Originality:** 3
**Rating:** 3
**Confidence:** 4

**Summary:**

This paper proposes a peer prediction-based mechanism to detect LLM-assisted cheating in crowdsourced annotation tasks where ground truth is unavailable. The core hypothesis posits that authentic human-generated responses exhibit distinct correlational patterns irreducible to LLM outputs. The introduced Conditioned Correlated Agreement (CAz) mechanism evaluates worker responses by quantifying inter-annotator correlations conditioned on requester-accessible LLM-generated signals (Z)
The paper addresses a timely problem with a theoretically grounded and empirically validated method. While assumptions require relaxation and scope expansion needs further validation, the CAz mechanism represents a significant advance in crowdsourcing data integrity preservation.

**Questions:**

How might CAz be adapted for high-dimensional, open-ended responses (e.g., peer reviews, creative writing) while maintaining theoretical guarantees? Does the embedding projection heuristic preserve information monotonicity properties?

**Ethical Concerns:**

["Major Concern: Human rights (including surveillance)"]

**Limitations:**

1. The major concern is still the blurring line between "Tool Use" and "Cheating": we are all becoming cyborgs in our daily work. At what point does using an LLM to assist with a task cross the line from acceptable tool use to data-corrupting cheating? Or, shouldn't we towards a more clear and inspired benchmarking that show the more interesting distribution that we want AI to be but not just reject something?
2. Is LLM-Polished Data bad? The paper doesn't provide a framework for navigating this gray area of legitimate augmentation versus outright replacement.
3. The risk of penalizing Neurodiversity and creative outliers. The method inherently rewards workers whose judgments align with the "diligent majority." What about a worker who is a genuine human but has a unique, outlier perspective? Or someone who interprets the task instructions in an unconventional but still valid way? Their responses might not correlate well with others, leading the CAz method to incorrectly flag them as low-effort or a "cheater." The system could inadvertently filter out cognitive diversity, which is often the very thing subjective tasks are trying to capture.

**Paper Formatting Concerns:**

Clarify notation consistency for $I_{\text{TVD}}$.

**Quality:**

2

**Strengths And Weaknesses:**

**Strengths**

1. Addresses the critical challenge of data integrity erosion due to undetected LLM usage in human feedback pipelines, particularly for tasks reliant on subjective human judgment (e.g., preference labeling, toxicity assessment).
2. The CAz mechanism provides a training-free, ground-truth-agnostic framework leveraging requester-generated LLM signals to disentangle genuine human effort from LLM-derived responses.

**Weaknesses**

1. The core requirement of this paper is that the principal can generate labels with an LLM $z$ to use for conditioning. This hinges on a crucial assumption: the principal knows which LLMs the cheaters are likely to use. If cheaters use a different proprietary model that the principal doesn't have access to, the effectiveness of the conditioning step is questionable. And
2. The strongest guarantees (for e-information monotonicity against any cheating strategy) rely on Assumption 4.1 and 4.2. However, the papers own empirical tests in Section 5.3 show that these assumptions "generally do not hold in practice". The authors then fall back on the weaker Assumption 4.5, which only provides guarantees against "naive" cheating strategies.
3. When the fraction of LLM cheaters is very low, the performance of the method degraded to traditional methods. The scene of this CAz can be clarified more specifically.
4. The further requirements of LLM data annotation would turn to more complex and open-ended text, which can exceed the tasks analyzed in this paper.

---

> ### Author Rebuttal · Authors · 2025-07-30
>
> We appreciate the reviewer for the thoughtful comments. Many of the concerns relate to the negative observations we made in the paper, and some generalizations of our method that we consider as future work. We would like to emphasize that **the goal of our paper is not to offer a panacea for the challenging problem of detecting LLM cheating WITHOUT ground truth, but rather to propose a useful solution and carefully examine when it works and when it does not.** However, we aim to show that the scenarios where our method works are fairly practical and important.
>
> Below, we respond to each point with more details.
> * Assumption on knowing the cheating LLMs
>   * **[This is a useful message]** We view the observation that $CA_Z$ is most effective when $Z$ is generated by the same model as $Z_i$ not as a weakness, but as a useful insight. It is an important warning to both the peer prediction and LLM communities: when conditioning on a mismatched LLM, the correlation among responses generated by the same LLM can exceed that of genuine human labels. In such cases, peer prediction mechanisms may fail to effectively incentivize or identify authentic human feedback.
>   * **[This is a practical assumption]** Our method is primarily designed for settings where a non-trivial fraction of users rely on the same popular LLM(s) or report uninformatively, in which case the principal can draw samples. This is a particularly harmful and practical setting, where existing baselines have limitations. In contrast, if only a few users adopt uncommon or proprietary models, their outputs tend to be less correlated and can be effectively distinguished from human labels by $CA_Z$, as well as by existing crowdsourcing algorithms. We show that $CA_Z$ performs competitively with the baselines across both settings, whereas each baseline fails significantly in at least one of them.
> * Weaker truthful guarantee
>   * **[This is a useful message]** We view the hardness of obtaining information monotonicity under arbitrary cheating strategies as a part of our contributions, not a weakness. As AI becomes better, it is a rather challenging task to fully distinguish LLM labels from human labels. Therefore, we consider it to be good news that we are able to offer a tool that is useful to address cheating with naive strategies.
>   * **[This is a practical assumption]** Furthermore, perhaps misled by its name, naive strategies are actually particularly important and common in practice. As highlighted in lines 136-140, agents tend to either decide to work for observing $X_i$ or decide to cheat and observe $Z_i$, but often never try to observe them at the same time. Any non-naive cheating requires simultaneously observing $X_i$ and $Z_i$ and thus is more like an adversarial behaviour rather than cheating.
> * Low fraction of LLM cheaters
>   * As noted above and in lines 354–357, the key advantage of $CA_Z$ lies in its theoretical guarantees and robust performance across settings. While knowing the fraction of LLM cheaters might allow for more tailored methods, this information is rarely available in practice—underscoring the value of a robust, general-purpose approach.
> * Generalization of $CA_Z$ to Open-ended crowdsourcing tasks
>   * Unfortunately, the heuristic generalization of $CA_Z$ (shown in Appendix F) does not technically preserve the truthful guarantees. The key issue is that the dimension of the embedding vector can be very high, making it difficult to directly estimate the mutual information between two embedding vectors. Therefore, we use cosine similarity as a heuristic alternative, which preserves the spirit of mutual information but is not technically an estimate of it.
>   * Even though the theoretical guarantee does not generalize, the performance suggests that the idea still works. However, further research on (weaker) theoretical guarantees for high-dimensional embeddings and advanced detection methods is beyond the purpose of the current paper.
> * Tool-using vs cheating, and LLM-polished data
>   * As mentioned in line 376, we view further distinguishing “tool use” and “cheating” as an important future work. For the main application considered in the current paper, i.e. subjective labeling tasks, we assume the principal is primarily interested in genuine human feedback, even though AIs (or human-AT teams) can do better in some aspects. In other words, we target the crowdsourcing tasks where AI usage is forbidden by the principal.
>   * Distinguishing between LLM-polished and LLM-generated responses is particularly relevant in tasks involving textual feedback, such as peer review, rather than in labeling tasks. We provide a discussion of the generalization of $CA_Z$ for textual responses in Appendix F, where the goal is exactly to distinguish responses based on the information they have rather than the vocabulary they use. However, as noted, our theoretical results do not directly extend to this setting, and we view this as a promising direction for future work.
> * Penalizing outliers
>   * We agree that outlier responses are less likely to receive high scores under peer prediction mechanisms. However, it is fundamentally difficult—if not impossible—to distinguish low-effort cheaters from genuine outliers based solely on observed responses. This tradeoff is not unique to $CA_Z$; it applies broadly to existing crowdsourcing algorithms. Due to this tradeoff, we can't enjoy the benefits without accepting the costs.
>   * Nonetheless, we are still designing new scoring mechanisms and are actively using them in practice. This is because a primary issue in crowdsourcing is that agents are not incentivized to exert effort, often resulting non-trivial fraction of data that might be pure noise. Therefore, we need scoring mechanisms to identify potentially unreliable responses and flag agents who may warrant further scrutiny. Importantly, **low scores do not imply automatic exclusion, but rather provide the principal with a tool to prioritize which data/agent may need additional attention.**

---

> > ### Comment · Reviewer_NyJA · 2025-08-07
> >
> > Thank you for your rebuttal comment. The paper's contribution within its stated assumptions is clear.
> >
> > However, I want to emphasize that the true frontier of this problem lies beyond these assumptions. The core challenge is not to perfect solutions for "naive" strategies, but to build theory and methods for the complex, "messy" reality of LLM use. **This is particularly critical because algorithms based on these simplified assumptions risk unfairly penalizing genuine human contributors, mistaking unique perspectives for statistical noise.**
> >
> > Future work should therefore prioritize the foundational challenges of: (1) Quantifying and distinguishing points on the spectrum of human-AI interaction, rather than relying on a binary cheating model. (2) Developing theory that is robust to a diverse and unknown landscape of LLMs.
> >
> > Continuing to build upon simplified assumptions will yield diminishing returns. I strongly encourage the authors to lead the way in confronting these more fundamental questions.
> >
> > I will keep my score.

---

> > > ### Author Response · Authors · 2025-08-07
> > >
> > > Thank you for the follow-up comments. We see your point that the "ideal" solution to LLM usage in crowdsourcing is to not only address the common naive strategies, but also any complex use of LLMs. **However, we believe it's worth pursuing useful solutions to a specific yet common setting, even if they are not universal.** We believe that progress in this area requires iterative development of models that can score and incentivize human effort.
> > >
> > > That said, we believe our method is a reasonable and useful solution to the problem we are considering.
> > > Below, we highlight a few points from our paper and rebuttal, and would greatly appreciate if the reviewer could elaborate on what an "ideal" or expected solution might look like in the setting we study.
> > >
> > > > "The core challenge is not to perfect solutions for "naive" strategies, but to build theory and methods for the complex, "messy" reality of LLM use."
> > >
> > > We fully agree that LLM usage in general settings can be messy. However, our focus is on a widely studied and practically relevant setting where **the principal only observes discrete labels from agents**. In this setting, we believe the "naive strategy" is a very reasonable space of low-effort behaviors, where agents can choose to work hard on an $\alpha$ fraction of tasks, report LLM labels on a $\beta$ fraction of tasks, and report uninformly on the remaining fraction of tasks for any $\alpha, beta$. More complex behaviors—such as using both the LLM and high-effort signals simultaneously—require more effort than truth-telling. **In our considered setting**, could the reviewer specify **why these assumptions are simplified, and what is "a diverse and unknown landscape of LLM usage" you are considering?**
> > >
> > > > "... rather than relying on a binary cheating model."
> > >
> > > We would like to clarify that our model is **not binary**. It explicitly allows agents to mix between high-effort, LLM-assisted, and random responses in arbitrary proportions (i.e., varying $\alpha$ and $\beta$).
> > >
> > > > "... risk unfairly penalizing genuine human contributors, mistaking unique perspectives for statistical noise."
> > >
> > > We appreciate this concern and have taken steps to address it. As pointed out by other reviewers and in our rebuttal, our method is intended for human-in-the-loop deployment. **It does not automatically discard any labels, but instead helps the principal flag responses that are more likely to be low-effort** (which could include LLM responses, random guesses, or even high-effort but atypical responses). These flagged responses can then be reviewed more carefully, rather than ignored outright. Could the reviewer specify **what the concern is for the human-in-the-loop implementation of our method?**
> > >
> > > We would also note that many widely used algorithms in crowdsourcing—such as variants of the Dawid-Skene model—carry similar risks of discounting minority or outlier responses, yet remain influential and widely applied. Could the reviewer clarify **whether the same concern applies to these baseline methods, and if not, why our method raises a qualitatively different risk?**

---

> > > > ### Comment · Reviewer_NyJA · 2025-08-07
> > > >
> > > > Thank you for the detailed rebuttal. The paper addresses the important problem of detecting LLM use in crowdsourcing and presents a mathematically detailed framework. The core contribution is the Conditioned Agreement ($CA_Z$) mechanism, designed to be robust against "naive" cheating strategies.
> > > >
> > > > However, my core concerns about the paper's simplified assumptions, fragile theoretical foundation, and the resulting fairness issues remain. The rebuttal defends these simplifications rather than addressing their significant implications.
> > > >
> > > > **1. The Model's Assumptions Do Not Capture Realistic "Low-Effort" Scenarios.**
> > > >
> > > > - **Example 1: The Rule-Based Cheater.** An agent could instruct an LLM:
> > > > 	```
> > > > 	Label any comment containing profanity as 'toxic', otherwise label it 'safe'. Create a multi-agents workflow.
> > > > 	```
> > > > 	Or even create an agent framework with tools like Claude-Code etc., for such task. This is a low-effort strategy, but resulting signal $Z_i$ is not a simple sample from the LLM's base  conditional distribution $P(\text{label}|\text{prompt})$; The paper's model, which relies on estimating a shared correlation structure, cannot account for this. That is only a simple case complex, "messy" reality of LLM use.
> > > >
> > > > *   **Example 2: The Justification Seeker.** In many platforms (like the Preference Alignment dataset cited), a justification is required, an agent might make a quick gut decision (e.g., A > B) and then prompt an LLM to write a plausible-sounding justification. The discrete label is human, but the signal a principal might check is AI-generated, poisoning the dataset in a way the model cannot detect.
> > > >
> > > > The framework's effectiveness is confined to the simplest forms of cheating, leaving it vulnerable to these and other more realistic strategies.
> > > >
> > > > **2. The Theoretical Foundation Itself is Ill-Suited for the Task.**
> > > >
> > > > The entire framework rests on using Total Variation Distance Mutual Information ($I_{TVD}$) as a proxy for truthful, high-effort work (e.g., in Assumption 4.5 and Proposition 4.6). This foundation is flawed for two reasons:
> > > >
> > > > *   **Conceptual Mismatch:** High mutual information ($I_{TVD}(X_i; X_j | Z)$) is a measure of *statistical correlation*, not truth or effort. Two workers who share a strong cultural bias will produce highly correlated answers, yielding a high score. A lone worker with a more objective, but dissenting, viewpoint will be penalized. The theory, by its very nature, rewards conformity and punishes potentially valuable, diverse perspectives.
> > > >
> > > > *   **Practical Breakdown:** In **Appendix F**, when confronted with high-dimensional text data, the method abandons its core theoretical measure (mutual information) and switches to a *heuristic* (cosine similarity). This is a tacit admission that the theoretical framework is not scalable or robust enough for complex, real-world data. Or turn to LLM-as-a-Judge, but this also raise new concerns, as some bench like Judgemark also showed the limitation of this way.
> > > >
> > > > The paper's extensive derivations create an illusion of rigor, but they build upon a foundation that is conceptually mismatched and practically brittle.
> > > >
> > > > **3. The Theoretical Flaws Directly Lead to Fairness and Ethical Concerns.**
> > > >
> > > > The model's reliance on rewarding statistical conformity is not just a technical weakness; it is an engine for generating bias. As noted, it systematically penalizes legitimate but minority viewpoints. My concerns on this front are consistent with the findings of the independent ethics review, which highlighted risks of "Cultural and Demographic Bias" and unfairly impacting worker "income and reputation." The "human-in-the-loop" defense is insufficient, as the algorithmic scores will inevitably be used to automate decisions at scale, amplifying this inherent bias.
> > > >
> > > > While the paper tackles an important problem, it builds an elaborate structure on a foundation that is ill-suited for the task. The chosen theoretical tool (mutual information) inherently rewards conformity, leading to a system that is neither robust to realistic cheating strategies nor fair to the human workers it evaluates.
> > > >
> > > > **My most fundamental concern is this: a method built upon such demonstrably weak assumptions should not be framed as a direct solution to a real-world problem as complex as annotator cheating.** To do so is to present a fragile tool as a robust one, risking the very real harm of penalizing fair work and suppressing the diverse human data our field relies upon.
> > > >
> > > > For these reasons, **I will keep my score.**

---

> ### Author Response · Authors · 2025-08-05
> **Thank you for your feedback**
>
> Dear reviewer NyJA,
>
> Thank you for acknowledging our rebuttal. We hope our responses have clarified your questions and concerns. If not, would you mind following up with additional questions or highlighting what assumptions/claims/results may require further effort? This feedback is essential for further improving our paper. Thanks!

---

> ### Author Response · Authors · 2025-08-08
>
> Dear Reviewer,
>
> Thank you very much for your thoughtful and detailed feedback—it has helped us better understand your concerns. Based on our reading, the main points raised are: (1) our method may not effectively distinguish genuine outliers from cheaters, and (2) our theoretical framework, which relies on correlations between reports, may not be well-suited to the annotator cheating problem.
>
> Before offering further clarifications, we would like to respectfully point out that **these concerns are not unique to our approach, but are shared by much of the existing literature on crowdsourcing quality control**, including [Dawid and Skene (1979), Wang et al. (2017), Zhou et al. (2012), Guo et al. (2023), Xia et al. (2024)], among others. Many of these are recent works published at NeurIPS and ICML. Like ours, these methods operate in settings without access to ground truth, and therefore rely on patterns of agreement or correlation among annotators to infer both label quality and annotator reliability.
>
> **While we fully acknowledge the limitations of such approaches, we also note that this framework remains widely adopted and actively developed in the field.** In this context, we hope it is understandable that our work—though applied to a new use case involving LLM usage—is built upon similar assumptions. We thus feel that a different standard of critique may not be entirely fair, simply because we frame LLM reliance as a form of "cheating."
>
> ---
> Below, we respond to your concerns at a lower level in a slightly different order.
>
> * The theoretical foundation, mutual information
>
> There may be some misunderstanding regarding the interpretation of mutual information in our framework. Importantly, mutual information does not penalize every agent who deviates from the majority. As shown in Line 178 and onward, our method learns a personalized delta tensor for each pair of agents, meaning that the scoring function $T_{\tilde{\Delta}}$ is agent-specific. This allows our model to capture structured deviations—such as when an agent $i$'s responses are negatively correlated with another agent $j$'s. In such cases, the scoring function will learn this pattern and assign agent $i$ a score significantly higher than that of a cheater under $CA_Z$.
>
> Furthermore, the peer prediction literature has established that **mutual information is a monotonic measure of effort or truthfulness** when genuine information is independent (there is no collusion on cheap signals) [Kong and Schoenebeck (2017), Shnayder et al. (2016)]. This property implies that only signals that contain information beyond what is already available (e.g., via LLM samples) will be rewarded. In your example 1, unless a large number of agents adopt the same LLM-based strategy, such cheating behavior will not yield high mutual information with human responses and will thus receive a low score.
>
> * Genuine outliers v.s. cheaters
>
> The main concern of the reviewer is that our method cannot effectively distinguish between these two types of agents, leading to potential ethical issues. However, in the setting we study—where only discrete agent reports are available and ground truth is absent—these two types of agents are **inherently indistinguishable**. **A cheater could always claim to be an outlier, and no unsupervised method can rule this out with certainty.** The only way to separate these two types of agents is via human verification, where our method can help prioritize these verifications by flagging them with lower scores. We are happy to add a warning for real-world applications of our method as a remark for ethical considerations.
>
> Moreover, we would like to emphasize that this challenge is not unique to our method. As mentioned, it is shared across the broader class of model-based methods in crowdsourcing quality control. Therefore, we respectfully suggest that **this limitation arises from the fundamental challenge of the lack of ground truth, rather than a shortcoming of our theoretical framework**. Instead, we view our theoretical understanding of the method as an advantage over prior work, not a weakness.
>
> * Example 2 and Appendix F
>
> Example 2 involves textual responses and thus falls outside the primary scope of our paper, which focuses on discrete labels. While our theoretical guarantees do not extend to the generalization of our method to textual responses (as discussed in Appendix F), we do not believe this suggests that the underlying theoretical framework is ill-suited to such settings. On the contrary, we show that although estimating mutual information directly becomes computationally intractable for high-dimensional textual data, the core idea—measuring information or similarity in addition to what is already captured by the LLM response—remains applicable and effective in practice.

---

> > ### Author Response · Authors · 2025-08-08
> > **Additional Reference**
> >
> > [Wang et al. (2010)] Jing Wang, Panagiotis G. Ipeirotis, Foster Provost. Cost-effective quality assurance in crowd labeling. 2017.
> >
> > [Zhou et al. (2012)] Dengyong Zhou, Sumit Basu, Yi Mao, John C. Platt. Learning from the wisdom of crowds by minimax entropy. 2012. NeurIPS
> >
> > [Guo et al. (2023)] Hui Guo, Boyu Wang, Grace Yi. Label Correction of Crowdsourced Noisy Annotations with an Instance-Dependent Noise Transition Model. 2023. NeurIPS
> >
> > [Xia et al. (2024)] Mingxuan Xia, Zenan Huang, Runze Wu, Gengyu Lyu, Junbo Zhao, Gang Chen, Haobo Wang. Unbiased Multi-Label Learning from Crowdsourced Annotations. 2024. ICML

---

### Official Review · Reviewer_CDQu · 2025-07-02

**Clarity:** 4
**Significance:** 4
**Originality:** 3
**Rating:** 5
**Confidence:** 3

**Summary:**

This paper develops a theoretically robust quality measure to evaluate the quality of human contributions in addition to existing LLM answers and detect workers with less informative responses.

**Questions:**

What is the computational cost of the score?

**Ethical Concerns:**

["NO or VERY MINOR ethics concerns only"]

**Final Justification:**

The paper is well-written with clear contribution

**Limitations:**

yes

**Quality:**

4

**Strengths And Weaknesses:**

Strengths:
- The paper studies an important task. The method is well-motivated and well-thought-out.
- The paper provided interesting insights regarding the pattern of human labelers and LLM labelers.
- The paper is well-written and clear.

The method is developed with a realistic setting and moderate assumptions, a good improvement over previous methods. The authors made it very clear about the scope and how to interpret the results, which I appreciate a lot. The writing is nice and easy to follow, with sufficient explanations and motivations. The experiments are well-designed and validate the claims made earlier.

Overall, I think it is a good paper.

---

> ### Author Rebuttal · Authors · 2025-07-30
>
> We’re grateful for the reviewer’s recognition of our contributions. For your question, the computational cost is very minor. Computing the scores once for the tested dataset takes less than 1 minute on a MacBook Pro (no repeating for computing error bars). We will add the computation requirement in the paper.

---

> ### Author Response · Authors · 2025-08-05
> **Thank you for your feedback**
>
> Thank you for your review and appreciation of our work. If you have anything further you'd like to discuss, we’d be happy to continue the conversation.

---

### Decision · Program_Chairs · 2025-09-17

**Decision:**

Accept (poster)

**Comment:**

This paper addresses the emerging problem of detecting LLM-dependent contamination in crowdsourcing.
A framework that works without ground truth is presented and both theoretical support and empirical effectiveness are shown.
The timeliness and novelty of the problem setting, as well as the convincing experimental results, are strong points that were appreciated by the reviewers.

At the same time, there are remaining concerns about the simplicity of the model and potential risks to fairness, such as unfair treatment of outliers or minority opinions. Ethical reviews also pointed out issues related to worker rights, privacy, and the unclear boundary between LLM assistance and cheating. The authors responded by promising improvements, including revising terminology, clarifying data usage, and emphasizing human-in-the-loop oversight.

Overall, the work makes a clear and timely contribution and points to an important direction for the field.